# Fast-backward replay of sequentially memorized items in humans

Qiaoli Huang[1,2,3†]*, Jianrong Jia[1,2,3,4†], Qiming Han[1,2,3,4], Huan Luo[1,2,3]*

[1]School of Psychological and Cognitive Sciences, Peking University, Beijing, China; [2]PKU-IDG/McGovern Institute for Brain Research, Peking University, Beijing, China; [3]Beijing Key Laboratory of Behavior and Mental Health, Peking University, Beijing, China; [4]Peking-Tsinghua Center for Life Sciences, Peking University, Beijing, China

**Abstract** Storing temporal sequences of events (i.e., sequence memory) is fundamental to many cognitive functions. However, it is unknown how the sequence order information is maintained and represented in working memory and its behavioral significance, particularly in human subjects. We recorded electroencephalography (EEG) in combination with a temporal response function (TRF) method to dissociate item-specific neuronal reactivations. We demonstrate that serially remembered items are successively reactivated during memory retention. The sequential replay displays two interesting properties compared to the actual sequence. First, the item-by-item reactivation is compressed within a 200 – 400 ms window, suggesting that external events are associated within a plasticity-relevant window to facilitate memory consolidation. Second, the replay is in a temporally reversed order and is strongly related to the recency effect in behavior. This fast-backward replay, previously revealed in rat hippocampus and demonstrated here in human cortical activities, might constitute a general neural mechanism for sequence memory and learning.

*For correspondence:
qiaolihuang@pku.edu.cn (QH);
huan.luo@pku.edu.cn (HL)

†These authors contributed equally to this work

Competing interests: The authors declare that no competing interests exist.

## Introduction

Storing and retrieving temporal sequences of events (i.e., sequence memory), a capacity shared across species, is crucial to many cognitive functions, including speech recognition, movement planning, and episodic memory (*Doyon et al., 2003*; *Giraud and Poeppel, 2012*). For example, we retrieve a stream of serially ordered events when recalling past personal experience, and we memorize numbers in order when calling a friend on the phone. Swinging the racket to hit an incoming ball in a tennis match similarly involves the planning and controlling of a chain of movement elements over time.

To accomplish sequence memory, two core components – the content (items) and the ordinal information (temporal order) – of a sequence are vital to be encoded and maintained in working memory. There exists ample evidence that neural responses during the retention interval show a sustained load-dependent enhancement (*Buschman et al., 2011*; *Jensen et al., 2002*; *Klimesch et al., 1999*; *Sauseng et al., 2009*; *Vogel and Machizawa, 2004*; *Xu and Chun, 2006*), suggesting that the maintenance of mnemonic contents is implemented by recurrent feedback loops (*Luck and Vogel, 2013*), as well as suppression of irrelevant information (*Sauseng et al., 2009*; *Bonnefond and Jensen, 2012*). That being said, retention of the sequence order information in memory cannot solely rely on an overall enhancement of neural activity and presumably requires the temporally segregated representations of individual items. Theoretical models postulate that sequence memory is mediated by a theta-gamma coupled neuronal oscillatory mechanism (*Jensen and Lisman, 2005*; *Lisman and Idiart, 1995*), such that individual items of the list/sequence, encoded in gamma-band activities, occur at the different phases of a theta-band rhythm. A recent MEG study provides important evidence supporting this theta-gamma coupling model during the

**eLife digest** Have you ever played the 'Memory Maze Challenge' game, or its predecessor from the 1980s, 'Simon'? Players must memorize a sequence of colored lights, and then reproduce the sequence by tapping the colors on a pad. The sequence becomes longer with each trial, making the task more and more difficult. One wrong response and the game is over.

Storing and retrieving sequences is key to many cognitive processes, from following speech to hitting a tennis ball to recalling what you did last week. Such tasks require memorizing the order in which items occur as well as the items themselves. But how do we hold this information in memory? Huang et al. reveal the answer by using scalp electrodes to record the brain activity of healthy volunteers as they memorize and then recall a sequence.

Memorizing, or encoding, each of the items in the sequence triggered a distinct pattern of brain activity. As the volunteers held the sequence in memory, their brains replayed these activity patterns one after the other. But this replay showed two non-intuitive features. First, it was speeded up relative to the original encoding. In fact, the brain compressed the entire sequence into about 200 to 400 milliseconds. Second, the brain replayed the sequence backwards. The activity pattern corresponding to the last item was replayed first, while that corresponding to the first item was replayed last. This 'fast-backward' replay may explain why we tend to recall items at the end of a list better than those in the middle, a phenomenon known as the recency effect.

The results of Huang et al. suggest that when we hold a list of items in memory, the brain does not replay the list in its original form, like an echo. Instead, the brain restructures and reorganizes the list, compressing and reversing it. This process, which is also seen in rodents, helps the brain to incorporate the list of items into existing neuronal networks for memory storage.

---

memory *encoding* period (*Heusser et al., 2016*). By asking subjects to mentally replay short sound or video clips, the temporal phase patterns of the neural activities during memory retrieval are found to be similar to that during memory encoding, further advocating preservation of temporal information of the stimulus in working memory (*Michelmann et al., 2016*). Working memory contents have also been suggested to be maintained in rapid transition in dynamic hidden states (*Stokes, 2015*; *Wolff et al., 2017*). However, it remains unknown how the human brain represents and maintains temporal order information during the memory *retention* period, when the stimulus sequence is not present.

Despite this gap, it is well-established in animal studies that place cells in rodent hippocampus fire in the same or reverse order in which they appeared during previous spatial navigation, in both awake and asleep status (*Diba and Buzsáki, 2007*; *Foster and Wilson, 2006*; *Louie and Wilson, 2001*; *Skaggs and McNaughton, 1996*). In contrast to rodent studies in which sequential trajectory of place cell firing could be explicitly assessed to explore reactivation profiles during memory maintenance, human studies have provided more limited access to the item-specific reactivation profiles at the neuronal ensemble level. To overcome this difficulty, in the present study, we employed sequence memory paradigms in combination with a temporal response function (TRF) method (*Crosse et al., 2016*; *Jia et al., 2017*; *Lalor et al., 2006*) to dissociate the item-specific neuronal response during the memory retention interval.

Our results demonstrate that each item in a memory list, characterized by an alpha-band neural profile, is successively reactivated to mediate the representation and preservation of ordinal information in memory. Importantly, this item-by-item neuronal profile does not simply echo the sequence in the external world but instead displays two intriguing properties compared to the actual stimulus sequence: reversal in order and compression in time. Specifically, the item that occupies a late position in the list is reactivated earlier (e.g., for sequence 1 – 2, the reactivation profile is 2 – 1 in order), and the whole list encompassing all items is compacted within a 200 – 400 ms temporal window. Most importantly, this backward reactivation profile is directly related to the subsequent recency effect in memory behavioral performance. Our study provides novel neural evidence in humans that the sequence order information is encoded and preserved in a fast and reverse reactivation sequence.

## Results

### Probing memory-related reactivations during retention period

Previous studies have provided evidence supporting the object-based nature of working memory such that all features of an object, even the task-irrelevant ones, will be stored automatically in working memory (*Gao et al., 2011*; *Hollingworth and Luck, 2009*; *Hyun et al., 2009*; *Luck and Vogel, 1997*). Most importantly, color features have been found to be the type of feature having the strongest conjunction with other features within an object (*Gao et al., 2011*; *Johnson et al., 2008*; *Wheeler and Treisman, 2002*). Based on these findings, in Experiment 1, we used task-irrelevant color probes that were either memory-related or non-memory-related to examine whether we could tag the memory reactivations during the retention period when the to-be-remembered features (i.e., orientation features) were not present.

We recorded 64-channel EEG signals from human participants performing an adapted working memory task (*Sawaki and Luck, 2011*). Each trial consisted of three periods: encoding, maintaining, and recalling (*Figure 1A*). In short, subjects were instructed to memorize the orientation of the cued bar ('encoding') and then held the information in working memory ('maintaining'), and finally performed a memory test on the orientation feature ('recalling'). During the maintaining period, participants performed a central fixation task (to control eye movements), with two task-irrelevant color discs presented simultaneously. Crucially, the two probes were either memory-related (i.e., with memory-matching color, WM) or non-memory-related (i.e., with memory-nonmatching color, NWM), and neither contained the to-be-memorized orientation features. They were displayed at either the left or right side of the fixation cross to exclude spatial memory effects.

Next, a TRF approach (*Jia et al., 2017*; *Lalor and Foxe, 2010*; *Ding and Simon, 2012*), which has been used to assess brain response that tracks ongoing changes in sound envelope (*Lalor and Foxe, 2010*; *Ding and Simon, 2012*), visual luminance (*Lalor et al., 2006*; *VanRullen and Macdonald, 2012*), and even high-level properties (*Liu et al., 2017*; *Broderick et al., 2018*), was employed to extract and isolate neuronal responses for the ongoing luminance change of the two probes (WM and NWM), respectively, throughout the maintaining period. Specifically, the luminance of the two discs was modulated continuously according to two 5 s random temporal sequences that were generated anew in each trial (*Figure 1B*). The TRF responses (*Figure 1C*) for the WM disc probe (WM-TRF) and NWM disc probe (NWM-TRF) were then calculated and dissociated from the same 5 s EEG recordings, based on their corresponding stimulus temporal sequences (*Figure 1B*). Note that the TRF response is defined as the brain response to a unit luminance in the stimulus sequence, as a function of temporal lag after each transient. As a result, the two calculated TRF responses (i.e., WM-TRF and NWM-TRF) would represent the impulse response for the luminance sequence of the WM and NWM probes, respectively, throughout the memory retention interval (see details of TRF approach in Materials and methods). If the color feature is not specifically bound to the associated orientation features held in memory, we would expect no difference between the WM-TRF and NWM-TRF responses. On the other hand, if the two probes show distinct responses, it suggests that the task-irrelevant color probes could be used to tag the memory-related activations during retention.

### Experiment 1: Alpha-band memory effects (one-item memory)

Eighteen participants participated in Experiment 1, and their memory performance was well above chance (N = 18, mean accuracy = 0.888, s.e. = 0.029; One-sample t-test, df = 17, $t$ = 26.34, p < 0.001, CI: [0.357, 0.419], Cohen's d = 6.208). First, the overall TRF response for the color probes showed clear alpha-band patterns and was robust at single-subject level (*Figure 2—figure supplement 1*), consistent with previous findings (*Jia et al., 2017*; *VanRullen and Macdonald, 2012*). We next performed a spectrotemporal analysis on the WM-TRF and NWM-TRF responses (see the corresponding TRF waveforms in *Figure 2—figure supplement 2A*) to examine their fine dynamic structures as a function of frequency (0 – 30 Hz) and time (0 – 0.6 s), and this was done on each channel and in each participant separately. As shown in *Figure 2A*, both WM-TRF (left panel) and NWM-TRF (right panel) demonstrated clear sustained alpha-band activations (8 – 11 Hz; permutation test, p < 0.05, corrected, see *Figure 2—figure supplement 2C*), consistent with previous findings (*VanRullen and Macdonald, 2012*). Given the prominent alpha-band activations in the TRF

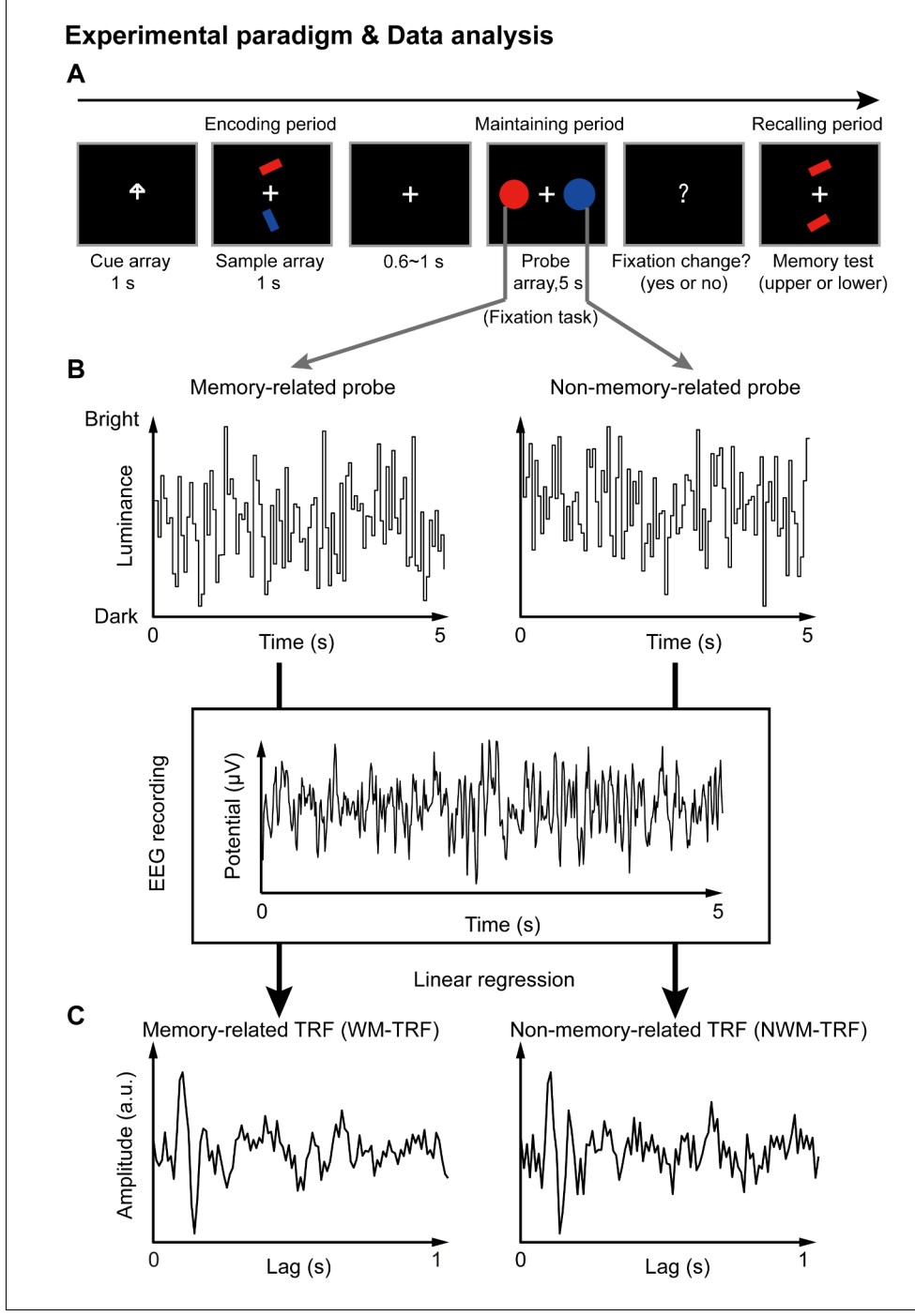

**Figure 1.** Paradigm for Experiment 1(1-item memory task) and illustration of the TRF approach. (**A**) A central arrow cue was first presented to indicate which of the bars the participants should examine. A sample array containing two bars then appeared and participants were asked to memorize the orientation of the cued bar ('encoding period'). Next, a probe array containing two task-irrelevant disc probes with memory-matching color and non-memory-matching color was presented for 5 s ('maintaining period'), during which participants performed a central fixation task by monitoring an abrupt luminance change of the central fixation cross, while simultaneously holding the orientation information in memory. In the final 'recalling period', participants were asked to judge which bar (upper or lower) had the same orientation as the cued bar. (**B**) During the 5 s maintaining period, the luminance of the two discs was continuously modulated according to two random temporal sequences respectively. At the same time, electroencephalography (EEG) responses were recorded. (**C**) The TRF approach was used to calculate the impulse brain response for the memory-related (left, WM-TRF) and non-memory-related
*Figure 1 continued on next page*

*Figure 1 continued*

(NWM-TRF) disc probes. TRF response characterizes the brain response to a unit increase in luminance of a stimulus sequence, as a function of time lag after each transient. Note that the WM-TRF and NWM-TRF were derived from the same EEG responses but were separated based on the corresponding stimulus luminance sequence. As a result, the computed WM-TRF and NWM-TRF would represent the elicited response for each unit transient at the WM and NWM probes, respectively, throughout the memory retention interval.

responses, we then first selected channels that showed overall significant larger WM + NWM alpha-band responses (one-sample t-test, p < 0.05; compared to the average of all channels). Among the resulting channels, three electrodes (red dots in *Figure 2B*) passed the statistical test on the WM-NWM alpha-band difference (bootstrap test, p < 0.06, see details in Materials and methods; FDR corrected across channels and time) and were defined as a 'channel-of-interest' for allsubsequent

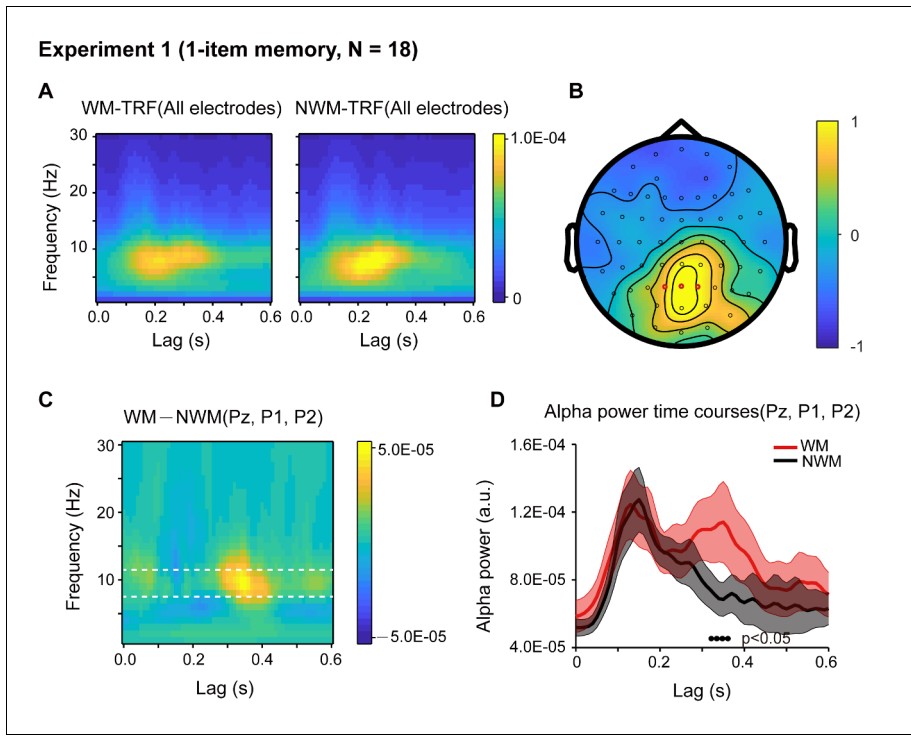

**Figure 2.** Results for Experiment 1 (1-item memory task). (**A**) Grand average (N = 18) time-frequency power profile for memory-related (WM, left) and non-memory-related (NWM, right) TRFs as a function of latency (0 – 0.6 s) and frequency (0 – 30 Hz) for all EEG channels. (**B**) Grand average distribution map for the normalized WM – NWM alpha-band (8 – 11 Hz) power difference in combination with the normalized WM + NWM alpha-band power within the first 0.6 s. Red dots indicate channel-of-interest (Pz, P1, P2) (see details of selection criteria in Materials and methods), which were analyzed for all subsequent analyses, including those in Exp. 2. (**C**) Grand average time–frequency plots for WM – NWM power difference as a function of latency and frequency in the channel-of-interest. (**D**) Grand average alpha (8 – 11 Hz) power time courses (mean ± SEM) for WM-TRF and NWM-TRF responses in the channel-of-interest. Black dots at the bottom indicate time points showing significant WM-NWM differences (bootstrap test, p < 0.05, FDR corrected across time points). See also *Figure 2—figure supplements 1–4*. The data are provided in source data 1.

The online version of this article includes the following source data and figure supplement(s) for figure 2:

**Source data 1.** Source data for *Figure 2*.
**Figure supplement 1.** Validation of TRF response.
**Figure supplement 2.** TRF waveforms in channel-of-interest (Pz, P1, P2) in Experiment 1 and Experiment 2 and the TRF power spectrum.
**Figure supplement 3.** Alpha-mediated memory effect did not show spatial lateralization, and analysis within other frequencies.
**Figure supplement 4.** Control analysis and eye movement distribution.

analyses, including those in Experiment 2. As shown in *Figure 2C*, a 'channel-of-interest' showed enhanced alpha-band power for WM-TRF compared to NWM-TRF within the latency of around 200 – 500 ms. We then extracted the alpha-band time courses within the channel-of-interest. As shown in *Figure 2D*, WM (red) showed significant alpha-band enhancement over NWM (black) from 310 ms to 370 ms (bootstrap test, p = 0.046, FDR corrected across time), supporting the alpha-band memory effect.

The WM and NWM probes were presented in two visual fields, and therefore the observed alpha memory effect could be a result of spatial attention (*Worden et al., 2000*). We performed two additional analyses to address this issue. First, we examined the alpha-band memory effects (WM – NWM) for the left and right discs separately and observed a similar spatial distribution without statistically significant lateralization effects (paired t-test, N = 18; see *Figure 2—figure supplement 3A*). Second, we compared the alpha power between channels that are contralateral to the WM probe and channels contralateral to the NWM probe and did not find any significant difference (paired t-test, N = 18; see *Figure 2—figure supplement 3B*). Thus, the alpha-mediated memory effects during maintenance could not be interpreted by spatial attention. This is also consistent with recent findings supporting memory-related hidden states in the absence of attention (*Wolff et al., 2017*).

It is noteworthy that during the central fixation task (see eye movement profile in *Figure 2—figure supplement 4B*), WM and NWM probes were both task-irrelevant, and the only difference between them regarded color (i.e., memory-related or non-memory-related). Therefore, the observed difference between the WM-TRF and NWM-TRF responses indicates that the task-irrelevant color probes did carry memory-related information, consistent with previous findings advocating the object-based nature of working memory (*Gao et al., 2011*; *Hollingworth and Luck, 2009*; *Hyun et al., 2009*). The results also suggest that we could use task-irrelevant color probes to tag the memory-associated reactivations during maintenance, using the alpha-band temporal profile as a neural signature (no effects in other frequency bands, *Figure 2—figure supplement 3C*). See further analysis on the alpha-band in TRF responses in *Figure 2—figure supplement 4A*.

## Experiment 2: Sequential alpha-band response for sequence memory

We next examined how a temporal sequence and the associated order information is maintained in working memory. Nineteen participants participated in Experiment 2. As shown in *Figure 3A*, participants were first presented with a sample array containing three bars with different orientations and different colors. Next, two of the three bars were randomly selected and serially cued, and importantly, participants were asked to memorize not only the orientation of the two cued bars but also their temporal order (i.e., the $1^{st}$ and the $2^{nd}$ orientation). During the following 5 s 'maintaining period', participants performed a central fixation task and held the orientation sequence in memory. In the final 'recalling period', participants compared the orientation of the bar in the test array with the memorized orientations (e.g. is it more similar to the orientation of the $1^{st}$ or the $2^{nd}$ cued bars?). Therefore, subjects should be memorizing the temporal order of the orientation sequence.

Crucially, the probe array during the maintaining period consisted of three disc probes – one $1^{st}$-memory-related probe (i.e., matching color with the $1^{st}$ cued bar), one $2^{nd}$-memory-related probe (i.e., matching color with the $2^{nd}$ cued bar), and one non-memory-related probe (i.e., matching color with the non-cued bar). The three discs were presented at three random spatial locations on a ring to exclude spatial memory effects. Again, the luminance of the three disc probes was modulated continuously according to three random sequences respectively. The corresponding TRF responses ($1^{st}$-WM-TRF, $2^{nd}$-WM-TRF, and NWM-TRF) were then computed from the same 5 s EEG recordings, and a spectrotemporal analysis was performed on them for each channel and in each participant separately (see TRF waveforms in *Figure 2—figure supplement 2B*). Notably, all the analysis was performed in the channel-of-interest, which was independently defined in Experiment 1 (see *Figure 2B*).

First, significant recency effect (i.e., $2^{nd}$ item better than $1^{st}$ item) in behavioral performance was found (N = 19, $1^{st}$ item: mean = 0.794, s.e. = 0.020; $2^{nd}$ item: mean = 0.840, s.e = 0.014; recency effect: mean = 0.047, s.e. = 0.020; paired t-test, df = 18, t = 2.378, p = 0.029, CI: [0.005, 0.089], Cohen's d = 0.532). Moreover, the alpha-band memory effects (i.e., WM – NWM difference averaged within the whole 600 ms time range, bootstrap test, p = 0.047) revealed in Experiment 1 were still present (*Figure 3B*). Most interestingly, the $1^{st}$-WM-TRF and the $2^{nd}$-WM-TRF displayed distinct alpha-band temporal courses (*Figure 3C*): the $1^{st}$-WM-TRF alpha-band response (red line) was

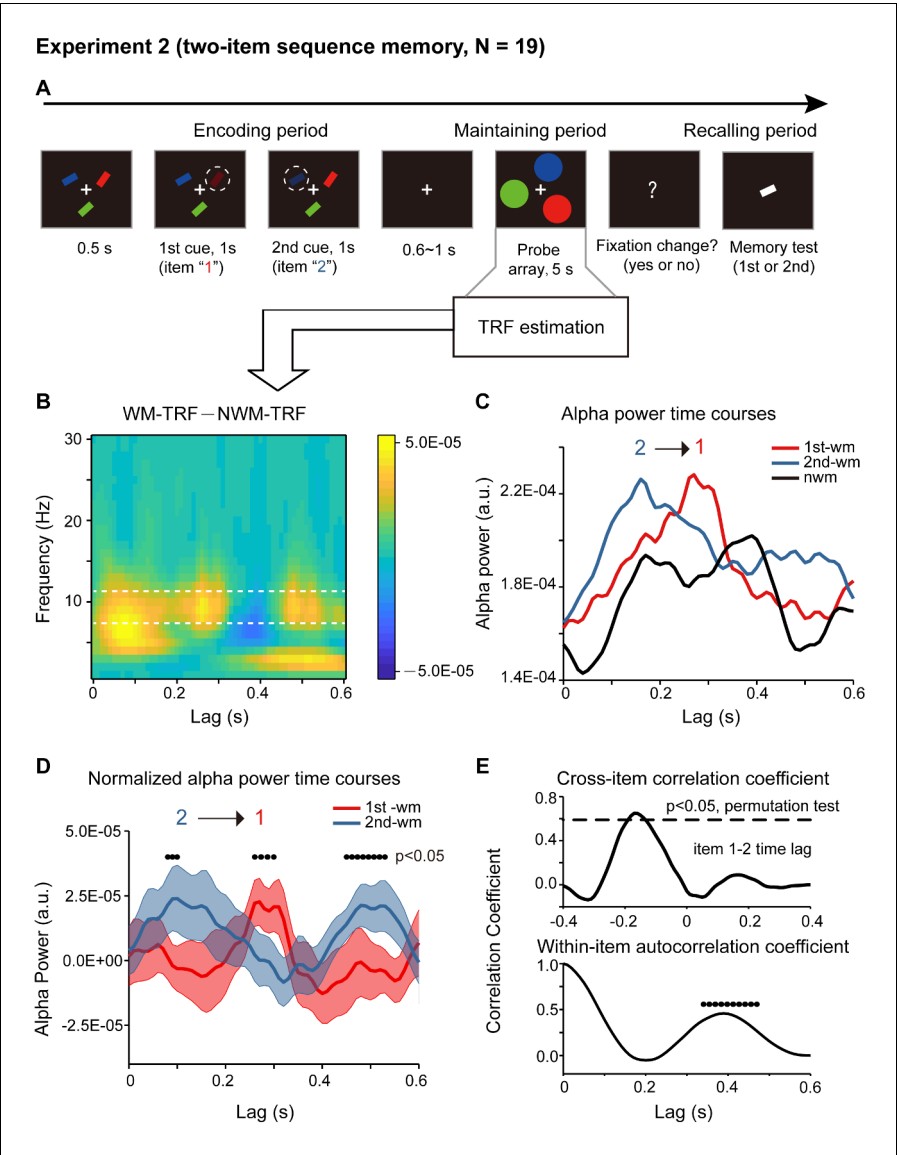

**Figure 3.** Paradigm and results for Experiment 2 (two-item sequence memory task) (**A**) Participants were first presented with a sample array containing three bars with different orientations and different colors. Next, two of the three bars were randomly selected and serially cued by dimmed luminance, and participants were asked to memorize not only the orientation of the two cued bars but also their temporal order (i.e., the 1st and the 2nd orientation). During the following 5 s 'maintaining period', participants performed a central fixation task and held the two-item orientation sequence in memory. In the final 'recalling period', participants compared the orientation of the bar in the test array with the memorized orientations (e.g. is it more similar to the orientation of the 1st or the 2nd cued bars?). (**B**) Grand average (N = 19) time-frequency power profile for the difference between WM-TRF (averaging across 1st-WM-TRF and 2nd-WM-TRF) and NWM-TRF responses in the channel-of-interest, which were independently defined in Experiment 1 (*Figure 2B*). (**C**) Grand average alpha (8 – 11 Hz) power time courses for 1st-WM-TRF (red), 2nd-WM-TRF (blue), and NWM-TRF (black) in the channel-of-interest. (**D**) Grand average normalized alpha power time courses (mean ± SEM) for 1st-WM-TRF (red) and 2nd-WM-TRF (blue). Black dots indicate points at which the alpha power was significantly above zero (p <0.05, one-sided paired t-test, uncorrected). (**E**) Top: cross-correlation coefficient between the normalized alpha-band profile for the 1st-WM-TRF and the 2nd-WM-TRF responses. The dashed lines represent the corresponding statistical significance threshold (p < 0.05, permutation test, corrected). Bottom: autocorrelation coefficient for the normalized alpha power time course (black dots: p < 0.05, permutation test, corrected). See also *Figure 3—figure supplements 1–2*. The data are provided in source data 2.

The online version of this article includes the following source data and figure supplement(s) for figure 3:

*Figure 3 continued*
**Source data 1.** Source data for *Figure 3*.
**Figure supplement 1.** Control analysis.
**Figure supplement 2.** Paradigm and results for three-item sequence memory task (N = 15).

temporally delayed by approximately 200 ms relative to that of the 2nd-WM-TRF (blue line) response (alpha-band peak latency, Wilcoxon signed-rank test, p = 0.036). We next normalized the alpha-band power temporal profiles by subtracting the averaged alpha-band time courses across all the three TRFs from that for 1st-WM-TRF, 2nd-WM-TRF, and NWM-TRF, respectively. *Figure 3D* illustrates the normalized alpha-band temporal profile, showing a reverse sequential activation profile (i. e., alpha-band activation for the 1st item followed by that for the 2nd item). Specifically, the 2nd-WM-TRF showed a response at around 80–100 ms, whereas the 1st-WM-TRF showed a response at around 260–300 ms (p < 0.05, one-sided t-test). The backward sequential response profiles were consistent across electrodes and items (see control analysis for each channel and for different item combinations in *Figure 3—figure supplement 1CD*).

To examine the statistical significance of the sequential activation, we calculated the cross-correlation coefficient between the normalized alpha-band profile for the 1st-WM-TRF and the 2nd-WM-TRF responses and then performed a permutation test by shuffling condition labels. As shown in *Figure 3E* (top panel), the 1st-to-2nd cross-correlation coefficient at a temporal lag of around –170 ms was significant (permutation test, p < 0.05, corrected), supporting that the 2nd item alpha-band activations essentially preceded those of the 1st item (i.e., negative temporal lag) by around 170 ms. Notably, the sequential activation pattern was still present for direct comparisons between the 1st-WM-TRF and 2nd-WM-TRF responses (*Figure 3—figure supplement 1A*; *Figure 3—figure supplement 1B*), supporting that it was not involvement of the NWM item (i.e., normalization) that caused the results. Furthermore, we observed a similar but weaker trend of reverse serial reactivation profile in a three-item sequence memory task (*Figure 3—figure supplement 2*).

Finally, in addition to the item-by-item sequential responses, we also observed a trend of repeated activations for the same item (*Figure 3D*). We thus calculated the within-item autocorrelation coefficient to examine the temporal period of the recurrent activations. As shown in *Figure 3E* (bottom panel), the temporal lag of ~390 ms was statistically significant (permutation test, shuffling between condition labels, p < 0.05, corrected). This suggests that the two items in the memory list are further organized into a recurring temporal chunk of approximately 400 ms, within which sequentially memorized items reactivate one after another in a reversed order.

## Relation to sequence memory behavior

We further evaluated whether the alpha-mediated memory reactivation during the retention period might have behavioral relevance. We first examined the relationship between the WM – NWM alpha-band enhancement and the one-item memory performance in Experiment 1. Participants were grouped based on their memory accuracy into 'high-performance' (N = 9, mean = 0.907, s. e. = 0.018) and 'low-performance' (N = 9, mean = 0.796, s.e. = 0.053) (*Figure 4A*). We then used the same analysis to assess the alpha-band courses for the two groups separately. Similar to the grand average results (*Figure 2D*), the high-performance group showed enhanced WM – NWM alpha-band power difference within 300–400 ms (*Figure 4B*), whereas the low-performance group showed weaker enhancement (*Figure 4C*). As shown in *Figure 4D*, the high- and low-performance groups displayed non-overlapping confidence intervals in the alpha-band memory effects (WM – NWM alpha power difference averaged from 310 to 370 ms, a time range selected based on group results, see *Figure 2D*). Thus, the stronger the alpha-band memory effects (i.e., Alpha $_{WM}$ > Alpha $_{NWM}$), the better the corresponding memory behavioral performances.

Participants in Experiment 2 (2-item sequence memory) were divided into two groups based on their memory recency effects (memory performance $_{2nd\ item}$ – memory performance $_{1st\ item}$): the 'high-recency' group (N = 9, mean = 0.116, s.e. = 0.020) and the 'low-recency' group (N = 9, mean = –0.021, s.e. = 0.017) (*Figure 4E*). First, both groups (*Figure 4FG*) showed similar reverse sequential responses profiles, similar to those in the group results (see *Figure 3D*). Notably, the 'high-recency' group displayed a strong and significant backward-sequential alpha profile (*Figure 4F*;

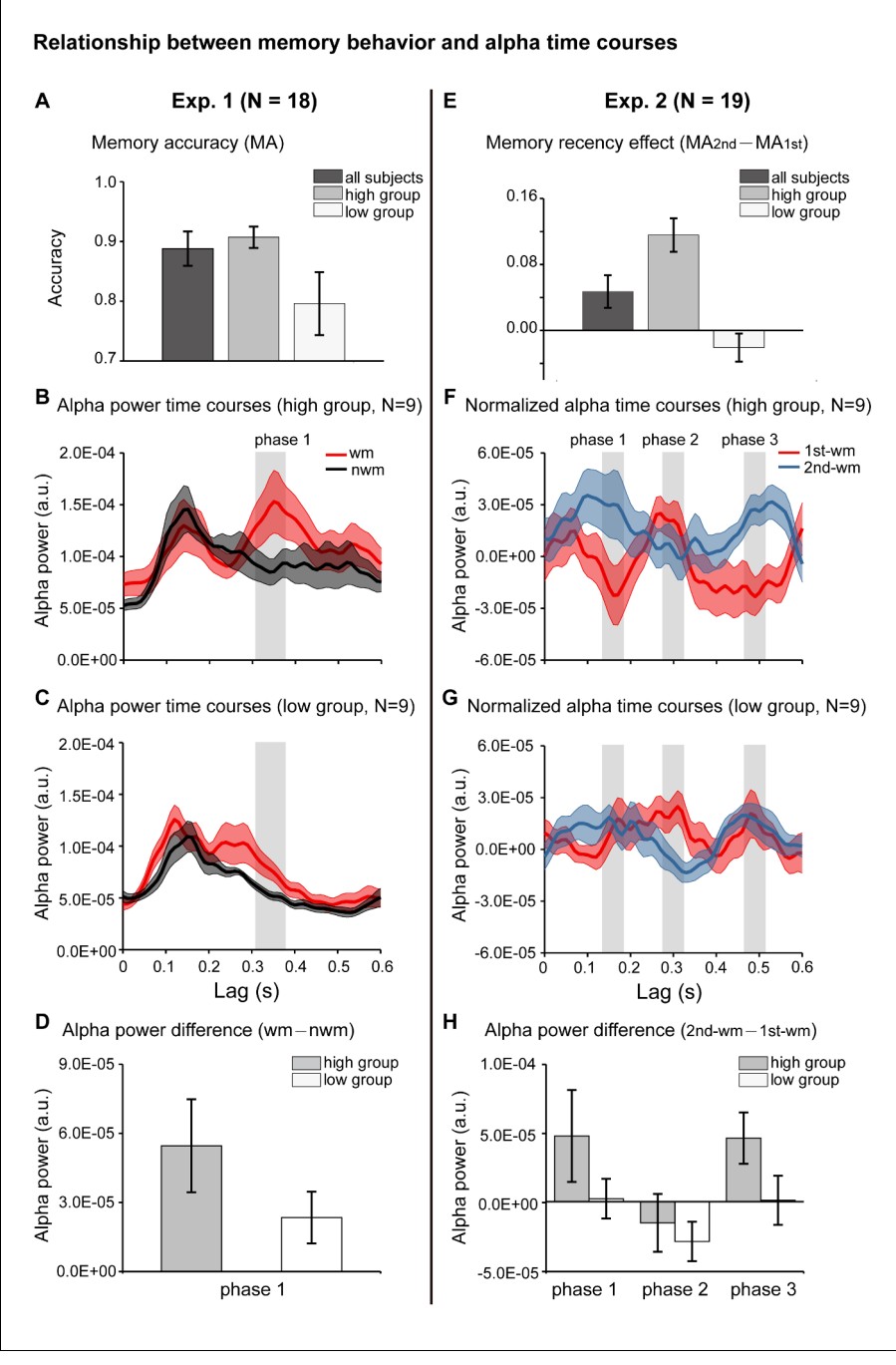

**Figure 4.** Relationship between memory behavior and alpha-band TRF profiles. (**A**) Participants in Experiment 1 were grouped based on the memory behavioral accuracy into 'high-performance' and 'low-performance'. Grand average (mean ± SEM) memory accuracy (MA) for all participants (N = 18), high-performance group (N = 9), and low-performance group (N = 9). (**B**) Grand average alpha-band power time courses (mean ±confidence interval) for the high-performance group (red: WM-TRF; black: NWN-TRF). (**C**) Grand average alpha-band power time courses (mean ±confidence interval) for the low-performance group (red: WM; black: NWN). The shaded box (310-370 ms) representing the time range for the alpha-band memory effect (WM – NWM), was defined based on group results (see *Figure 2D*). (**D**) Averaged alpha-band memory effect (WM – NWM) within the time range (shaded box in *Figure 4BC*) for the high- and low-performance groups (mean ±confidence interval). (**E**) Participants in Experiment 2 were grouped into two groups based on memory recency effects (MA for 2nd item –MA for 1st item). Grand average (mean ± SEM) memory recency effect for high-recency group (N = 9) and low-recency group (N = 9). (**F**) Grand average normalized alpha-band power time courses (mean ±confidence interval) for the high-recency group (red: 1st WM; blue: 2nd WM). The three shaded boxes (*phase 1*: 140 – 180 ms; *phase 2*: 280 – 320 ms; *phase 3*:

*Figure 4 continued on next page*

*Figure 4 continued*

470 – 510 ms), representing the time ranges for the sequential effect (2nd WM − 1st WM), were defined based on the group results (see *Figure 3D*). (G) Grand average normalized alpha-band power time courses (mean ±confidence interval) for the low-recency group (red: 1st WM; blue: 2nd WM). (H) 2nd − 1st alpha-band difference averaged within phase 1, phase 2, and phase 3 (shaded box in *Figure 4FG*) for the high-recency (grey) and low-recency (white) groups (mean ±confidence interval). The two groups showed non-overlapping confidence intervals for the 2nd −1st alpha-band difference in phase 1 and phase 3. All the confidence intervals were calculated using a jackknife procedure. See also *Figure 4—figure supplement 1*. The data are provided in source data 3.

The online version of this article includes the following source data and figure supplement(s) for figure 4:

**Source data 1.** Source data for *Figure 4* .

**Figure supplement 1.** Permutation test of the backward sequential reactivation for the high-recency and low-recency groups in Experiment 2.

---

permutation test, 1 − 2 cross-correlation coefficient at a lag of −170 ms, p < 0.05, *Figure 4—figure supplement 1A*), whereas the 'low-recency' group displayed weak and nonsignificant sequential responses (*Figure 4G*; permutation test, 1 − 2 cross-correlation coefficient at lag of −170 ms, p = 0.28; *Figure 4—figure supplement 1*). To further compare the difference in reactivation profiles between the two groups, we examined the 2nd-1st alpha power difference for the two groups at three time ranges (*phase 1*: 140 – 180 ms; *phase 2*: 280 – 320 ms; *phase 3*: 470 – 510 ms; shaded regions in *Figure 4FG*), which were selected based on group results (*Figure 3D*). As shown in *Figure 4H*, interestingly, the main difference between the high- and low-recency groups was on the 2nd TRF response (*Figure 4FGH*). Specifically, the high-recency group elicited a stronger 2nd item response than the low-recency group (phase 1 and phase 3, non-overlapping confidence intervals, *Figure 4H*; permutation test by shuffling between high- and low-recency groups, p = 0.08), whereas both groups elicited similar 1st item responses (phase 2, *Figure 4H*; permutation test by shuffling between high- and low-recency groups, p = 0.48). Thus, our results support an essential association between the backward sequential reactivations and recency effect in sequence memory behavior. We also performed an additional analysis by dividing subjects based on their memory accuracy rather than the recency, and the two groups elicited quite similar backward sequential responses (*Figure 3—figure supplement 1E*), excluding the possibility that it is memory strength rather than the recency effect that leads to the observed backward sequential responses.

Taken together, the alpha-mediated reactivation profile during the memory retention period is related to subsequent memory behavioral performance. Importantly, there exists an essential relationship between the backward item-by-item reactivation during maintenance and the subsequent recency effect, a behavioral index for sequence memory.

## Response bias control

In Experiment 2, subjects were instructed to decide whether the orientation of the bar in the test array was similar to the orientation of the 1st or the 2nd memorized orientations, and the behavioral performance showed a significant recency effect (i.e., 2nd item better than 1st item). Meanwhile, although the accuracy for both items was high, response bias might still be a confounding factor for the observed recency effect (e.g., biased to report the probe to better resemble the 2nd item). To address the issue, we ran a control study (N = 11) employing a new behavioral procedure that would not be contaminated by response bias. Specifically, the experiment paradigm was the same as that for Experiment 2 except that rather than making a binary response in the recalling period, subjects were instructed to rotate a bar to the memorized orientation. By calculating the angular deviation between the reported orientation and the true orientation of the memorized bar, in combination with a probabilistic model (*Bays et al., 2009*), we could estimate the target response probability for the 1st and the 2nd WM items separately, independent of response bias. As shown in *Figure 5A*, the target response probability was significantly larger for the 2nd item than for the 1st item (df = 10, one-tail paired t-test, $t$ = 2.052, p = 0.034, CI: [−0.003, 0.083], Cohen's d = 0.618), confirming the recency effect and excluding the response bias interpretation.

Furthermore, we combined the neural recordings in the control study (N = 11) with those from Experiment 2 (N = 19), resulting in a 30-subject dataset. As shown in *Figure 5B*, the normalized

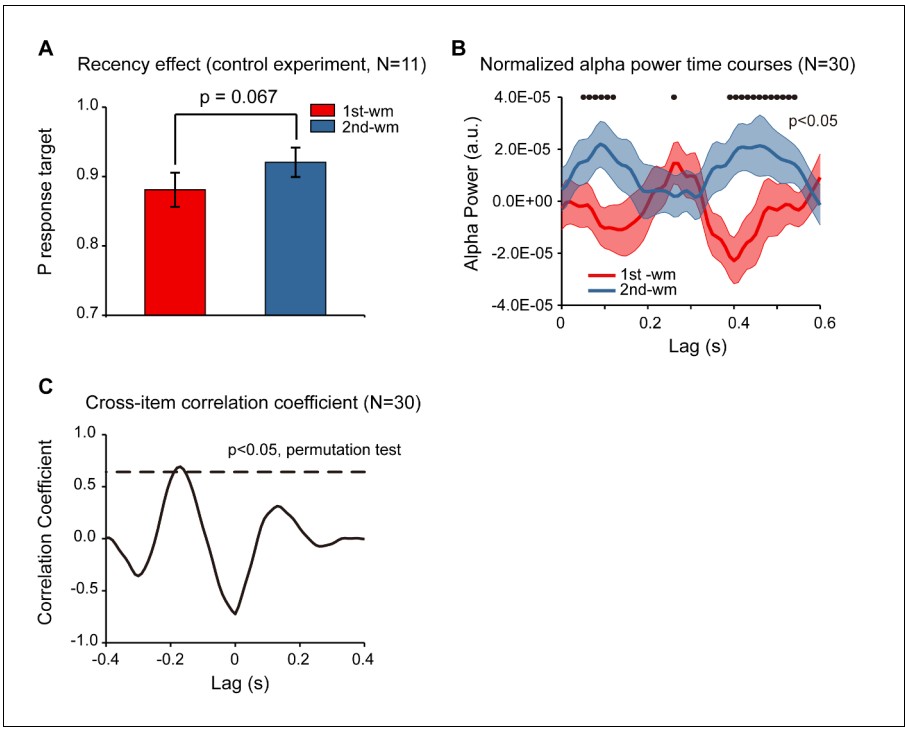

**Figure 5.** Response bias control and pooled results. (**A**) Grand average (mean ± SEM) target probability estimation in the control study (N = 11) for the 1st (red) and 2nd (blue) memorized orientation. The 2nd WM showed larger target probability than the 1st WM (*: p < 0.05, one-sided paired t-test), confirming the recency effect and excluding the response bias interpretation. Note that the procedure was the same as that for Experiment 2 except during the recalling period, subjects were instructed to rotate a bar to the memorized orientation rather than making a binary response. We next pooled the neural data from the control study and that from Experiment 2, resulting in a 30-subject dataset. (**B**) Grand average (N = 30) normalized alpha power time courses (mean ± SEM) for 1st-WM-TRF (red) and 2nd-WM-TRF (blue) of the pooled data. Black dots indicate points at which the alpha power was significantly above zero (p < 0.05, one-sided paired t-test, uncorrected). (**C**) Cross-correlation coefficient between the normalized alpha-band profile for the 1st-WM-TRF and the 2nd-WM-TRF responses of the pooled data. The dashed lines represent the corresponding statistical significance threshold (p < 0.05, permutation test, corrected). The data are provided in source data 4.

The online version of this article includes the following source data for figure 5:

**Source data 1.** Source data for *Figure 5*.

alpha-band reactivation profile (N = 30) showed a similar reverse sequential pattern. The 1st-to-2nd cross-correlation coefficient at a temporal lag of around −170 ms was significant (permutation test, p < 0.05, corrected; *Figure 5C*), again supporting that the 2nd item alpha-band activations essentially preceded those for the 1st item (i.e., negative temporal lag) by around 170 ms.

# Discussion

We recorded EEG responses in human participants performing sequence memory tasks and employed a TRF approach to probe the neuronal response that specifically tracks each item of the temporal sequence during memory retention. Our results consistently demonstrate that individual items are successively reactivated, characterized by a sequence of alpha-band activities. Compared to the actual stimulus sequence, the serial neural replay is temporally compacted within a 200–400 ms chunk and is reversed in temporal order. Crucially, the backward reactivation is associated with the recency effect, a behavioral index signifying backward memory priority. Taken together, our results constitute novel neural evidence in humans that the sequence order information is encoded and maintained in short-term memory by fast and backward serial reactivations.

Temporally sequenced activations have also been reported in several recent human studies. For example, after being trained in a reasoning task that involves selecting a statistically related object path, the brain spontaneously displays a fast successive representation of states (*Kurth-Nelson et al., 2016*). Recently, an fMRI study demonstrated that flashing only the starting point of a learned sequence would trigger a prediction wave of responses in primary visual cortices (*Ekman et al., 2017*). Finally, a MEG study examined the theta-gamma coupling strength as participants were presented with picture sequences, and found that the gamma power shifts along the phase of a theta rhythm as more items are added to memory, thus supporting the 'phase coding' model for episodic memory formation (*Heusser et al., 2016*). Our results are different from this important work and reveal distinct mechanisms. First, we examined the memory retention period when items were not presented, whereas the previous work studied the encoding period when items appeared serially. Second, we demonstrate backward reactivations whereas the previous study supports a forward-playing profile, further suggesting a fundamental distinction between memory encoding and retention.

We demonstrate that the item-by-item reactivation during memory retention period is temporally reversed in order. Previous neurophysiological recordings in monkeys have revealed forward serial activations in prefrontal population activity (*Siegel et al., 2009*). Sequences within the hippocampus in rats have been observed in both forward and reverse order (*Diba and Buzsáki, 2007*; *Foster and Wilson, 2006*; *Louie and Wilson, 2001*; *Skaggs and McNaughton, 1996*), and interestingly, the backward sequence has been shown during awake periods immediately after spatial experience (*Foster and Wilson, 2006*). A recent MEG work found a backward trajectory of representations of states (*Kurth-Nelson et al., 2016*) during a reasoning task. Why is the sequential reactivation reversed in temporal order during memory maintenance? One interpretation comes from a reinforcement learning model, which proposes that the recent item would be the starting and anchoring point for propagating information backward along incoming trajectories to build a recency-weighted running average of the rewards received for each action taken (*Foster and Wilson, 2006*; *Bornstein et al., 2017*; *Gershman and Daw, 2017*). Another interesting possibility is derived from the 'activity-silent' model (*Stokes, 2015*), which proposes that the contents of working memory are mediated by dynamic hidden states. These memory-related hidden states could be pinged and inferred out during maintenance in the absence of attention and lingering delay activity (*Wolff et al., 2017*). In our experiments, the TRF response could be regarded as the emerging item-specific activation after each pinging (i.e., each unit luminance transient in the probe stimulus). Consequently, the item associated with stronger excitability (i.e., the recent item) would appear with shorter latency compared to the item associated with weaker excitability (i.e., the early item), resulting in reversed sequential reactivation as we observed here. Furthermore, instead of presenting each item sequentially (*Siegel et al., 2009*), here all the items in the list were displayed simultaneously followed by serial cuing (to exclude the possibility that serial presentation might result in different encoding strengths). How the sequence is presented (sequential or simultaneous) could be another factor accounting for the order of reactivations.

Memory reactivations during the rehearsal period are compressed in time, in agreement with previous neurophysiological recordings in animals (*Foster and Wilson, 2006*; *Euston et al., 2007*; *Ji and Wilson, 2007*; *Skaggs et al., 1996*; *Xu et al., 2012*). Reorganization of events with regard to the internal temporal framework, presumably mediated by neuronal oscillations of various rhythms (e.g., theta, alpha, gamma, etc.), may contribute to memory consolidation. Discrete epochs of events that are temporally separated in the external world can be successfully associated in the brain within an appropriate time frame, which could be vital for operation of the neuronal plasticity mechanism (*Jensen and Lisman, 2005*; *Nabavi et al., 2014*), and a similar time compression mechanism could mediate sequence maintenance in working memory.

A recent EEG study revealed that attention samples multiple visual objects in a temporally sequential manner even when subjects maintained sustained attention on one object (*Jia et al., 2017*). We would argue that sequential attentional sampling could not account for the present observations. First, a previous study showed that attention would first sample the item associated with highest attentional priority and then would shift to an item associated with less attentional priority. However, here, all items in the sequence should be attended to with the same priority and thus would not be sequentially sampled by attention. Second, a previous sequential pattern (attended before unattended) could not reasonably account for the backward reactivation.

Furthermore, the backward sequential pattern and its association with the recency effect could be explained by a 'memory strength' interpretation. For example, the better the memory strength is of the item (e.g., better memory strength for the 2nd item over the 1st item), the earlier would be the response latency (e.g., earlier response for the 2nd item compared to the 1st item). To test the account, we performed a control analysis by grouping subjects in Experiment 2 into two groups based on their memory accuracy rather than the recency, and examined the respective response profiles. If the alternative interpretation were correct, we would expect to see a temporal shift in the response pattern between the two groups such that the high-accuracy group would show earlier response compared to the low-accuracy group. However, the two groups elicited quite similar backward sequential responses (*Figure 3—figure supplement 1E*), excluding the possibility that it is memory strength rather than a recency effect that leads to the observed backward sequential responses

Finally, our results exhibit alpha-mediated responses during memory retention, consistent with previous findings on working memory (*Jensen et al., 2002*). It is well acknowledged that alpha-band neuronal activities have a key function in working memory by efficiently suppressing distracting information (*Bonnefond and Jensen, 2012*). For instance, increased memory load is associated with enhanced alpha-band activities (*Jensen et al., 2002*; *Klimesch et al., 1999*; *Tuladhar et al., 2007*), and repeated transcranial magnetic stimulation (TMS) at alpha frequency modulates the short-term memory capacity (*Sauseng et al., 2009*). Memory reinstatement has been found to be accompanied by alpha-band activities, which carry a temporal phase signature of the replayed stimulus (*Michelmann et al., 2016*). In addition to the traditionally posited inhibitory function in many cognitive processes (*Sauseng et al., 2009*; *Bonnefond and Jensen, 2012*; *Klimesch, 2012*; *Händel et al., 2011*), alpha-band neuronal oscillations have recently been found to implement an ''echo'' or reverberation of the inputs. These alpha echoes are indeed enhanced by visual attention, supporting an additional role of the alpha-band rhythm in the maintenance of sensory representations over time (*VanRullen and Macdonald, 2012*). Taken together, we reason that the memory-related alpha response in our results reflects memory reactivations rather than inhibition, based on several facts. First, previous studies using a TRF approach have revealed enhancement in alpha-echoes under attended conditions (*Jia et al., 2017*; *VanRullen and Macdonald, 2012*), which could not be interpreted by inhibitory account. Second, the memory-related alpha responses did not show spatial attention effects (see *Figure 2—figure supplement 3A*), which are known to be mediated by inhibitory alpha activities (*Worden et al., 2000*). Finally, WM showed stronger alpha responses than NWM here and were strongly associated with memory behavior, further excluding the inhibitory interpretation. We propose that the alpha-band activity in TRF responses might represent the stimulus-induced phase resetting of the intrinsic alpha-band oscillation in EEG activities (*Makeig et al., 2002*), given their close associations (*Figure 2—figure supplement 4A*).

In conclusion, our results provide converging evidence that sequence order information is represented and consolidated in short-term memory by an item-by-item serial reactivation mechanism. Crucially, the serial reactivation, temporally compressed (i.e., within a 200 – 400 ms window) and reversed in temporal order (i.e., backward reactivation), is essentially associated with the subsequent memory behavioral performance. This fast-backward replay during maintenance, previously revealed in rat hippocampus for spatial navigation and shown here in humans for a non-spatial task, might therefore constitute a general neural mechanism for sequence memory and learning.

## Materials and methods

### Participants

Twenty subjects participated in Experiment 1. Twenty subjects participated in Experiment 2. Eleven subjects participated in the control study. During data collection and preprocessing, two subjects in Experiment 1, and one subject in Experiment 2 were excluded because of overall low behavioral performance, excessive eye movements, or not finishing the whole experiment. All participants had normal or corrected-to-normal vision and had no history of psychiatric or neurological disorders. All experiments were carried out in accordance with the Declaration of Helsinki. All participants provided written informed consent prior to the start of the experiment, which was approved by the Research Ethics Committee at Peking University (2015-03-05c2).

## Stimuli and tasks

Participants sat in a dark room in front of a CRT monitor with 100 Hz refresh rate, and their heads were stabilized using a chin rest. In each trial of all experiments, there were three periods: the encoding period, maintaining period, and recalling period (*Figure 1A*). Participants were asked to only memorize the orientation of the cued bars (0.38°×1.15° visual angle).

During each trial, all participants were instructed to keep the number of eye blinks to a minimum. Eye movements were monitored using an EyeLink 1000 eye tracker (SR Research). The experimental trial was initiated only when fixation was located within 1° visual angle of the fixation cross. Results showed that the participants maintained good fixation on the central cross (within 1°) throughout each trial (*Figure 2—figure supplement 4B*).

### Experiment 1 (one-item memory)

In Experiment 1, participants were asked to memorize the orientation of the cued bar (*Figure 1A*). At the beginning of each trial, a central arrow cue (1 s duration) pointing to upper or lower locations was presented to indicate which of the following bars the participants should examine. A sample array containing two bars then appeared at 3° visual angle above and below the fixation for 1 s. One bar was red and the other was blue. The orientation of one bar was chosen randomly within a range between 25° and 65° and the other was between 115° and 155° (0° is vertical). The colors and orientations at each location were counterbalanced across trials. Participants were instructed to memorize the orientation of the bar at the cued location ('encoding period'). After a blank interval (0.6 ~ 1 s), a probe array containing two task-irrelevant discs (3° in radius) with either memory-matching color or non-memory-matching color were presented to the left or right of the fixation (7° in eccentricity) for 5 s ('maintaining period'). The colors of the two disc probes were counterbalanced across trials. During this 5 s maintaining period, participants performed a central fixation task by continuously monitoring an abrupt luminance change of the central fixation cross, while simultaneously holding the orientation information of the cued bar in memory. In 25% of trials, luminance of the fixation cross would dim for 0.2 s during 1 s to 3.8 s, and participants were asked to make a self-paced response afterward. The purpose of this fixation task was to control eye movements during memory maintenance, given the presence of two luminance-changing color probes. In the final 'recalling period', a test array consisting of two bars appeared at 3° visual angle above and below the fixation. One bar was in the same orientation as the memorized bar, whereas the other was tilted +20° or −20° relative to the memorized orientation. The locations (upper or lower) of the matching and nonmatching bars were counterbalanced across trials. Participants would judge which bar (upper or lower) had the same orientation as that of the cued bar. Participants completed 128 trials in total.

### Experiment 2 (two-item sequence memory)

Experiment 2 was a two-item sequence memory task. Different from Experiment 1, participants were asked to memorize not only the orientations of two serially cued bars but also their temporal order (*Figure 3A*). At the beginning of each trial, three bars with different orientations and different colors (red, green or blue) were presented at the same eccentricity to the fixation (3° visual angle). Next, two of the three bars were randomly selected and serially cued by dimmed luminance to inform participants to memorize their orientations and temporal order (orientation of the 1st cued bar and the orientation of the 2nd cued bar). The orientation of the first memorized bar was randomly selected between 30° and 150°. The second memorized bar was tilted +25° or −25° relative to the first memorized bar. The non-memorized bar was tilted +12.5° or −12.5° relative to the first memorized bar. Next, three disc probes with the same colors as that of the three bars were presented for 5 s. The locations of three probes were randomly selected on a ring with a radius of 7° to exclude spatial memory effects ('maintaining period'). During this period, participants were asked to do a fixation task by continuously detecting whether the luminance of fixation cross changed. At the end of each trial, a test bar was presented at the center of the screen with orientation tilted +5° or −5° relative to the 1st memorized or 2nd memorized bar. Participants should compare the orientation of the bar with the memorized orientations (e.g. is it more similar to the orientation of the 1st or the 2nd cued bars?). Participants completed 96 trials in total.

### Control study (two-item sequence memory)

In the control study, all procedures were the same as in Experiment 2 except that during the recalling period, rather than making a binary response, subjects were instructed to rotate a test bar to either the 1st memorized orientation or the 2nd memorized orientation, and the orientations of the two bars were randomly selected between 0° and 180°. By calculating the angular deviation between the reported orientation and the true orientation of the memorized bar, in combination with a probabilistic model (*Bays et al., 2009*), we could estimate the target response probability for the 1st and the 2nd WM item separately, independent of response bias.

### Luminance modulation

A TRF approach was employed to examine and dissociate the probe-specific impulse responses during the 5 s memory maintaining period. To enable the TRF estimation, the luminance of each of the disc probes was independently modulated in time at each refreshing frame (100 Hz monitor refresh rate), according to the corresponding temporal sequence that was generated anew in each trial (two temporal sequences in Experiment 1; three temporal sequences in Experiment 2). Each random sequence was tailored to have equal power at all frequencies by normalizing the amplitudes of its Fourier components before applying an inverse Fourier transform. Luminance sequences ranged from dark (0 cd/m$^2$) to bright (13 cd/m$^2$).

## EEG recordings

EEG was recorded continuously using a 64-channel EasyCap and two BrainAmp amplifiers (Brain-Products). Vertical and horizontal electrooculograms were recorded by two additional electrodes around the participants' eyes. EEG data were offline band-pass filtered between 2 and 50 Hz. Independent component analysis was performed independently for each subject to remove eye-movement and artifact components, and the remaining components were back-projected onto the EEG electrode space. All channels were then referenced to the average value of all channels. The EEG was downsampled to 100 Hz before TRF calculation to be matched with the temporal resolution of the luminance sequences. To avoid the influence of the onset EEG response in each trial, which may bias the estimated TRF results, we extracted the middle part of the 5 s EEG trial responses (0.5 – 4.5 s) for further TRF calculation.

## Data analysis

### TRF computation

The mapping between the stimulus luminance sequence and the recorded EEG data was characterized by TRF response using the multivariate temporal response function (mTRF) toolbox (*Crosse et al., 2016*; *Lalor et al., 2006*). The TRF response describes the brain's response for unit transient in the stimulus temporal sequence and characterizes the brain's linear transformation of a stimulus input, S(t), to the neural response output, R(t), as R(t) = TRF*S(t), where * denotes the convolution operator. Specifically, the TRF response for each probe (e.g., 1st WM TRF, 2nd WM TRF, and NWM TRF), as a function of temporal lag, was calculated from the same EEG recordings, based on the corresponding stimulus temporal sequences (*Figure 1B*). Specifically in the present experiment, the resulted TRF responses would characterize the neural response for each unit transient of the luminance sequence at each probe (e.g., 1st WM, 2nd WM, and NWM TRF) throughout the memory retention interval. In each experiment, the TRF analysis was performed for each probe, on each channel, and in each subject respectively.

The TRF analysis, based on the linear relationships between the luminance sequence and the brain response, has been previously used and validated in both visual and auditory studies to examine the brain response that specifically tracks sound envelope or luminance sequence (*Jia et al., 2017*; *Lalor and Foxe, 2010*; *Ding and Simon, 2012*; *VanRullen and Macdonald, 2012*). Recently, TRF has even been revealed to track high-level features, such as global form coherence (*Liu et al., 2017*) and semantic dissimilarity (*Broderick et al., 2018*). In fact, the TRF approach is quite a robust and conservative way to assess the neural response to luminance change. Previous studies have also demonstrated that applying a quadratic extension of the linear TRF approach only resulted in slight improvement compared to a linear model (*Lalor et al., 2008*; *Power et al., 2011*), further advocating the linear assumption in TRF calculation.

Reliable estimation of TRF response requires neural signals with enough temporal length as well as the corresponding length of stimulus sequences, from which the TRF response characterizing their linear relationship could be calculated. Previous study has proposed that neural signals of at least 120 s are needed to obtain TRF responses with reasonable SNR (*Handy, 2009*). Theoretically, dissociating multiple TRFs from brain signals would require even longer signals. Back to our studies, each trial was 5 s, and we only analyzed the middle 4 s to avoid influences from onset and offset response. Therefore, to obtain TRF responses for multiple probes (e.g., two TRFs in Experiment 1 and three TRFs in Experiment 2), we would need at least 40 trials for the calculation. Therefore, TRF calculation at single-trial level, unless the trial is designed to be very long, could not meet the signal length requirement.

## Time-frequency analysis

The obtained TRF responses were then analyzed with MATLAB (MathWorks, Inc., Natick, Massachusetts), using the FieldTrip toolbox (*Oostenveld et al., 2011*) and wavelet toolbox functions to examine the spectrotemporal power profiles. Specifically, the TRF temporal profile was transformed using the continuous complex Gaussian wavelet (order = 4; for example, FWHM = 1.32 s for 1 Hz wavelet) transform (Wavelet toolbox, MATLAB), with frequencies ranging from1 to 30 Hz in increments of 1 Hz, for each probe, on each channel, and in each subject respectively. The alpha-band (8 – 11 Hz) power profiles were then extracted from the output of the wavelet transform for further analysis.

## Channel-of-interest

Given the prominent alpha-band activations in the TRF responses, we first selected channels that showed overall significant larger WM + NWM alpha-band TRF responses (one-sample t-test, two-tailed, $p < 0.05$; compared to the average of all channels) within the 0.6 s in Experiment 1. Among the resultant 17 channels (P7, P5, P3, P1, Pz, P2, P4, P6, P8, PO7, PO3, POz, PO4, PO8, O1, Oz, O2), we next searched those showing significant memory effect (difference between WM and NWM conditions) using a bootstrap procedure with multiple comparison correction. Three parietal electrodes (Pz, P1, and P2) passed the test and were selected as a 'channel-of-interest' for all subsequent analyses, including those in Exp. 2 and the control study. The alpha-band power profiles were then averaged within the channel-of-interest for all subsequent analyses. It is notable that two channels (Pz, P2) passed the test with $p \leq 0.05$, and three channels (Pz, P1, P2) passed the test for $p \leq 0.06$ (marginally significant). We then set the p threshold as 0.06 to incorporate three channels to increase the signal-to-noise ratio. Furthermore, selection first based on the overall EEG alpha power led to the same channel selection.

## Normalization

The alpha-band power profiles of the TRF responses were normalized by subtracting the averaged alpha-band time courses across all the TRFs (e.g., WM and NWM in Experiment 1; 1st WM, 2nd WM, and NWM in Experiment 2) from each alpha-band TRF profile and in each subject respectively. It is notable that in Experiment 1, given there was only one WM item, the normalization WM – (WM +- NWM)/2 was equal to (WM – NWM)/2. We were more interested in examining the item-specific activation profile relative to the general TRF response (e.g., how item one was different from all the other probes), and therefore chose the averaged alpha-band profile across all probes rather than that of the NWM probe only.

## Correlation coefficient calculation

To examine the sequential activation in Experiment 2, we calculated the cross-correlation coefficient between the normalized alpha-band profile for the 1st-WM-TRF and the 2nd-WM-TRF responses (Experiment 2).

## Bootstrapping test

A bootstrap procedure with multiple comparison correction was used to search channels-of-interest showing memory effects (i.e., WM – NWM alpha-band difference). Specifically, for each of the 17 channels showing prominent alpha-band response, we randomly resampled the WM – NWM alpha-band activities with replacement 18 times for the 18 participants in Experiment 1 and computed the

means of the estimated measures for the resampled participants. This procedure was repeated 1000 times. As a result, the distribution of the resampled WM – NWM alpha difference data was obtained and was compared to 0 to calculate the corresponding p-value for each time point and each channel. Next, we used FDR to correct the multiple comparison across channels (17 channels) and time (0 – 600 ms).

To examine the time range for the WM – NWM alpha-band difference within the channel-of-interest in Experiment 1, a bootstrapping procedure (bootstrapping number = 1000) was used to resample the WM – NWM alpha-band difference time course with replacement and the corresponding p-value was calculated for each time point. Next, a FDR was used to correct the multiple comparison across time. To examine the overall WM – NWM alpha-band difference within the channel-of-interest in Experiment 2, a bootstrapping procedure (bootstrapping number = 1000) was used to resample the WM – NWM alpha-band difference averaged over the first 600 ms with replacement, and the corresponding p-value was calculated.

## Permutation test

To examine the spectral contents of the TRF responses across all channels in Experiment 1, we did a permutation test by shuffling the relationship between stimulus sequence and EEG signals 500 times, from which the 0.05 threshold level as a function of frequency was estimated. Multiple comparison was corrected by setting the maximum threshold across frequencies as the corrected threshold.

To test the statistical significance of the sequential activations (cross-correlation coefficient) in Experiment 2, we performed a permutation test by shuffling the labels across conditions (i.e., across $1^{st}$-WM-related, $2^{nd}$-WM-related, and NWM-related probes). This shuffling was performed in each subject separately. We then used the same analysis for the original data to recalculate the normalized TRF responses and the corresponding cross-correlation coefficients as a function of time lag. The analysis was repeated 500 times, and a distribution for the cross-correlation coefficient at each time lag was obtained, from which the 0.05 permutation threshold was estimated. The largest threshold across all temporal lags was set as the multiple-comparison corrected threshold (corrected). The same permutation was performed to test the within-item autocorrelation coefficients. To test the statistical significance of the sequential activations (cross-correlation coefficient) in three-item sequence memory, we performed a permutation test by shuffling the labels across conditions (i.e., across $1^{st}$-WM-related, $2^{nd}$-WM-related, $3^{nd}$-WM-related, and NWM-related probes). To test the statistical significance of the sequential activations in the raw data (unnormalized responses) in Experiment 2, we performed a permutation test by shuffling the labels across the $1^{st}$-WM-related and $2^{nd}$-WM-related conditions.

As to the statistical test for the cross-correlation coefficient in the high-recency and low-recency subject groups in Experiment 2, we did the same permutation test within each group respectively. To compare the difference between groups, we performed a permutation test by shuffling data between high and low groups for 500 times. The original high-low group difference (the averaged $2^{nd} - 1^{st}$ alpha power in phase 1 and phase 3) was then compared to the permutation distribution, resulting in its corresponding p-value.

## Confidence interval estimation

Considering the small sample size of each group (e.g., high- and low-performance groups in Experiment 1; high- and low-recency groups in Experiment 2), we used a jackknife procedure (*Efron and Tibshirani, 1993*), during which one subject was removed at a time, to estimate the confidence interval of the mean of the alpha-band responses in each group (e.g., high- and low-performance groups in Experiment 1; high- and low-recency groups in Experiment 2).

## Time range selection

In Experiment 1, the time range (*phase 1*: 310 – 370 ms), defined based on group results (N = 18, *Figure 2D*), was used to examine the memory effects (WM – NWM alpha-band difference) for the high- and low-performance groups. In Experiment 2, the three time ranges (*phase 1*: 140 – 180 ms; *phase 2*: 280 – 320 ms; *phase 3*: 470 – 510 ms), defined based on group results (N = 19, *Figure 3D*),

were used to examine the sequential reactivation profiles (2nd WM- 1st WM alpha-band difference) for the high- and low-recency groups.

## Acknowledgements

We thank Dr. David Poeppel, Dr. Weiwei Zhang, Dr. Fang Fang, Dr. Lucia Melloni, Dr. Zaifeng Gao, Dr. Ling Liu, and Dr. Nai Ding for helpful comments. This work was supported by National Natural Science Foundation of China Grants 31522027 and 31571115 to LH.

## Additional information

### Funding

| Funder | Grant reference number | Author |
| --- | --- | --- |
| National Natural Science Foundation of China | 31522027 | Huan Luo |
| National Natural Science Foundation of China | 31571115 | Huan Luo |

The funders had no role in study design, data collection and interpretation, or the decision to submit the work for publication.

### Author contributions

Qiaoli Huang, Conceptualization, Data curation, Formal analysis, Validation, Investigation, Visualization, Methodology, Writing—original draft; Jianrong Jia, Data curation, Formal analysis, Investigation, Methodology, Writing—original draft; Qiming Han, Formal analysis, Methodology; Huan Luo, Conceptualization, Resources, Supervision, Validation, Visualization, Writing—original draft, Project administration

### Author ORCIDs

Qiaoli Huang (iD) http://orcid.org/0000-0003-4592-9270
Jianrong Jia (iD) https://orcid.org/0000-0001-7665-1182
Huan Luo (iD) http://orcid.org/0000-0002-8349-9796

### Ethics

Human subjects: All participants provided written informed consent prior to the start of the experiment, which was approved by the Research Ethics Committee at Peking University (2015-03-05c2).

### Decision letter and Author response

Decision letter https://doi.org/10.7554/eLife.35164.sa1
Author response https://doi.org/10.7554/eLife.35164.sa2

## Additional files

### Supplementary files

• Transparent reporting form

### Data availability

All data generated during this study are included in the manuscript and supporting files.

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
