## [Decision Letter]

Thank you for submitting your article "Fast-backward replay of sequentially memorized items in humans" for consideration by *eLife*. Your article has been reviewed by three peer reviewers, and the evaluation has been overseen by a Reviewing Editor and Michael Frank as the Senior Editor. The following individual involved in review of your submission has agreed to reveal her identity: Saskia Haegens (Reviewer #3).

The reviewers have discussed the reviews with one another and the Reviewing Editor has drafted this decision to help you prepare a revised submission.

This paper addresses an important question about how multi-item sequences are maintained in the brain. The paper is clear and well written and the analyses and results are novel. However, the paper rests on an assumption about probe tagging and WM maintenance, for which the case is not convincingly made in the paper. Furthermore, the bulk of the data is from group data. In order to be considered further, data from individual trials should be shown and analyzed.

For a revision to be successful, you must address the following three major issues and I am appending a list of other comments below that you can use to improve the paper as well:

1) All reviewers either did not follow the logic of the tagging idea or believed it was based on a number of assumptions that were not clearly demonstrated in the paper. The interpretation of the primary conclusions of this paper rest on the assumption that the TRF method can reliably 'tag' unique activation patterns for distinct stimuli. However, there is concern that a number of steps/assumptions that must be true in order for the author's interpretations of the results to be sound. First, there must be a relationship between the neural representation of the items during encoding and the neural representation of just the color feature of the item during maintenance, and the unique representations of these item/features must be dissociable. Second, there must be a linear relationship between the modulation of the luminance of the disc and the neural representation of the color of the disc. Third, the temporal dynamics of alpha power in the resulting TRF must serve as a unique proxy for the neural representation of each item. In order to draw the conclusions, it seems to me that each of these must be true. Please provide a more thorough explanation of the logic and assumptions behind using the TRF method to tag item reactivations and evidence that these assumptions are reasonable and supported by the data.

2) All reviewers felt strongly trial-by-trial evidence for item maintenance should be presented. For the reported patterns to truly reflect a physiological pattern akin to replay, it would be important to demonstrate that individual electrodes/subjects sequentially represent each of the two different stimuli on individual trials. Otherwise, there is the concern that the results may be the effect of averaging such that the observed patterns were caused by different items, electrodes, or subjects showing activations at different latencies according to list position. I think it is vital that a true replay analysis be performed and the authors need to explain this analysis very clearly. The authors can satisfy this concern by ensuring that they report and document an analysis that is focused on identifying replay at the single trial level.

3) Could the timing differences between the item reactivations be completely explained by the recency effect, or differences in memory strength by encoding position? For example, if we manipulated the sequence such that the item in the first position was always more salient than the item in the second position, would the temporal coding pattern during the maintenance period also flip (item 1 reactivated first, item 2 reactivated second)? A finding like this would suggest that the temporal reactivation differences are less about the encoding of order and more about the strength of encoding, or the 'attentional priority' of the items.

[Editors’ note: this article was subsequently rejected after discussions between the reviewers, but the authors were invited to resubmit after an appeal against the decision.]

Thank you for submitting your work entitled "Fast-backward replay of sequentially memorized items in humans" for consideration by *eLife*. Your article has been reviewed by three peer reviewers, and the evaluation has been overseen by a Reviewing Editor and a Senior Editor. The following individuals involved in review of your submission have agreed to reveal their identity: Andrew Heusser (Reviewer #1); Saskia Haegens (Reviewer #3); Rasa Gulbinaite (Reviewer #4).

Our decision has been reached after consultation between the reviewers. Based on these discussions and the individual reviews below, we regret to inform you that your work cannot be considered further for publication in *eLife* at this time.

The reviewers all agreed that the revision was very responsive and served to increase confidence in the results as well as clarified the methodological approach. However, there were still significant questions about the validity of the TRF approach especially given that the methods and analyses are so complex. Thus, as you can see, we reached out to another reviewer, an expert in the method. The comments of this reviewer echoed the concerns of the other reviewers and highlight the important need to validate the method with chromatic stimuli as suggested. Furthermore, for the reasons highlighted, this needs to be done on a single subject level. Because this would require new data collection and analyses, it is not clear how these analyses will turn out. However, if you are able to validate the paradigm with chromatic stimuli in the manner suggested by reviewer 4, as well as address all of the other concerns reviewer 4 brings up, we could consider a new submission at that time.

We have appended the reviews below but please note that reviewers 1 and 3 made their comments as an initial response to the revision which was very responsive. However it was upon further consultation with these reviewers that it was clear we needed to consult with an expert on the TRF methods.

*Reviewer #1:*

The authors did an excellent job at responding to my concerns and I have no additional major concerns.

*Reviewer #3:*

I believe the authors adequately addressed most of the concerns raised. The only remaining issue is whether these results hold on a single-trial level. While I understand that the TRF (by definition) cannot be computed for single trials, I would have liked to see an intermediate solution: recompute the analysis for binned trials (i.e., smaller subsets). However, I do understand the SNR concerns raised, and it may have to remain for future work. Maybe the authors can at least add a couple sentences to the Discussion to that extent?

Related to this, I did not understand how Figure 3D proves consistency over subjects? Because it is normalized? Maybe I am missing the point, but plotting the single subject results in addition to the grand average would have been more compelling.

*Reviewer #4:*

I have read the manuscript by Huang et al., the reviewing editor's summary of the reviewers' comments, and the authors' response to these comments. Although the authors extensively revised the manuscript based on the reviewer's comments, I agree with my predecessors that the assumptions that TRF approach allows to track memory activation of distinct stimuli is not sufficiently validated.

To my knowledge, there has not been yet a single paper validating the TRF approach for color stimuli (only achromatic stimuli have been used by several groups so far, e.g. Lalor and VanRullen), thus the authors first need to provide evidence that color stimuli can elicit a reliable TRF. This can be done by presenting a single colored stimulus and comparing the observed TRF with that obtained by shuffling the pairing between luminance sequences and concurrently recorded EEG time series. Crucially, this has to be demonstrated on a single-subject level. The latter is particularly important, because periodicity of TRF for achromatic stimuli has been shown to differ across participants, and the peak frequency of TRF is highly correlated with individual alpha peak frequency (VanRullen and Macdonald, 2012). Thus, group average analyses of TRF time courses (as done here), unless demonstrated that between-subject variability in TRF was not significant, can be driven by the subjects that have higher prominent TRF amplitude.

Based on the example TRFs in Figure 1, TRFs for chromatic stimuli differ substantially from the previously reported TRFs for achromatic stimuli (e.g. VanRullen and Macdonald, 2012), which reverberate up to 1 sec. Moreover, given the previous reports on difference in alpha-band power for different color stimuli (e.g. Yoto et al., 2007; DOI: 10.2114/jpa2.26.373), it is likely that TRF for different color stimuli would also be different. However, it is an empirical question that needs to be addressed prior to using TRFs for chromatic stimuli to tag items held in memory.

It has been previously reported that the phases of two TRFs elicited by achromatic flickering stimuli on the left and right hemifield exhibit phase differences across many channels (Lozano-Doldevilla and VanRullen, DOI: https://doi.org/10.1101/190595). Thus, it is possible that the phase differences reported here are not the property of the items stored in memory, but rather reflects the differences in propagation of different TRFs across cortex. Thus, again it is important to validate the TRF approach for several chromatic stimuli and to demonstrate that the phase differences are not related to the location of the stimuli in the retinotopic space.

Removing grand-average TRF responses from condition-specific TRF responses defeats the purpose of subject- and condition-specific TRF (subsection “Normalization”), and more importantly creates artificial oscillatory pattern (Figure 3D).

The electrode selection procedure is arguably biased to find condition differences. The authors selected the electrodes of interest based on the significant differences in TRFs between the WM and NWM conditions (subsection "Channel-of-interest", as well as authors' response: "According to the suggestion, we first selected channels based on the EEG alpha power, from which we then seek channels showing significant memory effects in TRFs."), and then show that there are indeed differences in TRFs in all subsequent analyses. To be statistically appropriate, channel selection procedure cannot involve data from condition differences, and should only be selected based on condition average or other criteria that are orthogonal to the subsequent analyses.

Overall, despite the interesting topic and clever experimental approach taken by the authors, the manuscript in its current form does not feel up to the empirical and analytic standards one typically associates with *eLife*.

[Editors’ note: what now follows is the decision letter after the authors submitted for further consideration.]

Thank you for choosing to send your work entitled "Fast-backward replay of sequentially memorized items in humans" for consideration at *eLife*. Your letter of appeal has been considered by a Senior Editor and a Reviewing Editor, and we are prepared to consider a revised submission with no guarantees of acceptance.

The appeal provided additional useful information that increased the confidence in your reported effects. However, there were still issues brought up by the reviewers that were misunderstood and/or not clearly addressed. These issues (outlined below) must be addressed in full in a revision in order for the paper to be acceptable either with data or added discussion.

1) The additional information you provided about the counterbalancing of color and location was very useful. An additional control that would strengthen your argument would be to directly compare the preferred phase of the TRF for different color/location combinations. The appeal included an additional figure illustrating that there is no effect of location in the alpha power timeseries. If possible with the existing data, it would useful to see the same analysis by color. If there are no reliable differences in the alpha power time course by color, then this cannot be a likely explanation for the phase differences for the different items maintained in memory.

2) Based on the new data, and re-analysis of another dataset, chromatic TRFs seem to be indeed oscillatory and comparable to the previously reported achromatic TRFs at the single-subject level. This is reassuring and adds credibility to the use of white-noise chromatic stimuli for tagging memory items. However, no information was included about the number of trials used in this control study to derive TFRs? Is it comparable to what has been used in the original study? Please add this information into the paper.

3) Also, it is not clear whether the plots provided in the appeal letter come from a single-color stimulus (which color?) or average of 3 colors? Please make this clear in the paper. Thus, this leaves open the original question asked by reviewers – are different color TRFs comparable in amplitude and phase (the property of TRFs that the results of this study hinge on)? Please show the data for each color separately.

4) Several statistical decisions remain unclear and need to be clearly motivated and described. For example, the paper stated "In Experiment 1, the "channel-of-interest" has passed statistical test after multiple-comparison correction across time and channels”. In other words, we used a non-biased approach (statistical test after multiple comparison correction) to determine channels showing memory effects ("channel of interest")." Thus, it appears that a multi-step procedure was used to select the channels of interest? The logic for this is not explained. Please explain the logic carefully and clearly. Although seemingly minor, why did you choose a p value of p =.058 instead of the conventional.05? (In the previous version reviewers also noted that details of a 'bootstrap' or 'permutation' test were also not clearly described). If you choose to submit a revised manuscript, please ensure that *all* statistics are clearly explained, motivated and reported. Otherwise, we cannot consider the paper suitable for publication.

5) A question was raised about whether the observed sequential replay was due to the normalization procedure. In the appeal, you noted 'First, we could see clear sequential activation pattern in the raw data (Figure 3C)." However, there is a concern that the normalization procedure itself (subtracting the average of the 3 conditions) seems to introduce an artificial oscillatory pattern. Looking at Figure 3C (non-normalized data), non-working memory item is arguably the only one that is oscillatory (most likely noise), and by removing the average of the three-curves one would end up with an oscillatory pattern in Figure 3D – the figure that summarizes the main finding of the paper. Please address this issue more directly by assessing whether there is indeed any evidence for oscillations in the raw data – without the normalization procedure.

---

## [Author Response]

This paper addresses an important question about how multi-item sequences are maintained in the brain. The paper is clear and well written and the analyses and results are novel. However, the paper rests on an assumption about probe tagging and WM maintenance, for which the case is not convincingly made in the paper. Furthermore, the bulk of the data is from group data. In order to be considered further, data from individual trials should be shown and analyzed.For a revision to be successful, you must address the following three major issues and I am appending a list of other comments below that you can use to improve the paper as well:1) All reviewers either did not follow the logic of the tagging idea or believed it was based on a number of assumptions that were not clearly demonstrated in the paper. The interpretation of the primary conclusions of this paper rest on the assumption that the TRF method can reliably 'tag' unique activation patterns for distinct stimuli. However, there is concern that a number of steps/assumptions that must be true in order for the author's interpretations of the results to be sound.

Thanks for raising the crucial concern and we apologize for failing to make the underlying logic and assumptions clear in the previous manuscript. According to the suggestions, we have provided a more thorough explanation of all the logic and assumption as below, and have added more clarifications and references in main texts (see details as below). We have also performed new control analysis (see Figure 2—figure supplement 1C and Figure 2—figure supplement 2C).

First, there must be a relationship between the neural representation of the items during encoding and the neural representation of just the color feature of the item during maintenance, and the unique representations of these item/features must be dissociable.

As pointed out by the reviewer, we probed the neural representation of the memorized items by tagging the neural response of the associated color feature. This assumption is actually based on previous findings and Experiment 1 was actually designed to test this hypothesis. Specifically, previous studies have provided evidence supporting the object-based nature of visual working memory such that all features of an object, even the task-irrelevant features, will be stored automatically in working memory (Luck and Vogel, 1997; Hollingworth & Luck, 2009; Hyun et al., 2009; Shen et al., JEP: HPP, 2015). Most importantly, color feature has been found to be the type of features that have the strongest conjunction with other features within an object (Johnson, Hollingworth and Luck, 2008; Wheeler and Treisman, 2002; Shen et al., JEP: HPP, 2015). Based on these findings, we chose the color feature to probe the neural representation of memory-related information during maintenance. It is noteworthy that we do not propose that any features could be employed to probe memory reactivations; indeed, we made use of the strong conjunction characteristic of color features to probe the memory-associated activations here. We have added more rationales and references in the Results (subsection “Probing memory-related reactivations during retention period”, first paragraph).

Furthermore and most crucially, we did not take the assumption for granted and indeed ran Experiment 1 to test the idea by examining whether the neural response for the task-irrelevant color probe during maintenance would be modulated by whether or not it was associated with memorized orientation. If the color feature is not bound to the corresponding memorized feature, we would expect no difference in response for the memory-associated and non-memory-associated color probes. Crucially, we did find the difference (Figure 2), supporting that the task-irrelevant color probe carries memory-related information and could be used to tag the neural representation of memorized features. We have added more clarifications in the manuscript (subsection “Probing memory-related reactivations during retention period”, last paragraph; subsection “Experiment 1: Alpha-band memory effects (one-item memory)”, last paragraph).

As to the concern about unique dissociable representations of the item/features, it is exactly our motivation to employ TRF approach so that we could dissociate brain response specifically for different probes from the same EEG recordings. In particular, because the luminance of each probe was modulated independently according to a random temporal sequence, we could calculate the TRF response for each probe based on its specific luminance sequence. We have added more clarifications in texts (subsection “Probing memory-related reactivations during retention period”, last paragraph).

Second, there must be a linear relationship between the modulation of the luminance of the disc and the neural representation of the color of the disc.

The reviewer is correct that TRF response is calculated in terms of the linear relationship between stimulus luminance and neural response. In fact, TRF approach has been used and validated in previous visual and auditory studies, which assessed brain response that specifically tracks sound envelope or luminance sequence (Lalor and Foxe, 2010; Ding and Simon, 2012; VanRullen and Macdonald, 2012; Jia et al., 2017). Luminance tracking is actually one of the most robust findings in visual studies. Recently, the TRF has even be revealed to be able to track high-level features, such as global form coherence (Liu et al., 2017) and semantic dissimilarity (Broderick et al., 2018). Therefore, TRF approach would be a quite robust and conservative way to assess the neural response to luminance change. Furthermore, previous studies have also shown that applying a quadratic extension of the linear TRF approach only resulted in slight improvement compared to linear model (Lalor et al., 2008; Power et al., 2011), further advocating the linear assumption for TRF method. To make the points more clear, we have added more references and clarifications about the linear assumptions and validations in Results (subsection “Probing memory-related reactivations during retention period”, last paragraph) and Materials and methods (subsection “TRF computation”, second paragraph).

Nonetheless, we reanalyzed our data to assess the linear concern in our results. Specifically, we divided the whole luminance sequences into three quartiles (low-luminance, middle-luminance, high-luminance), and recalculated their corresponding TRF responses separately. As shown in Author response image 1, the power of the TRF responses showed a decreasing pattern as luminance decreased, again supporting the trend of linear relationship between stimulus luminance and neuronal responses.

Third, the temporal dynamics of alpha power in the resulting TRF must serve as a unique proxy for the neural representation of each item.

Again, we are sorry for not making the alpha-band points clear in our previous version. In fact, using the alpha-band profiles to tag neural representation of each item derives from the results in Experiment 1. The rationales are as follows. First, the TRF responses for both probes and across all EEG channels showed prominent alpha-band responses (see Figure 2A). We next performed a permutation analysis to examine the spectral contents of the TRF responses and the alpha-band showed significant activations (Figure 2—figure supplement 2C), consistent with previous findings using TRF approach (VanRullen and Macdonald, 2012; Jia et al., 2017). Second, only the alpha-band activity showed memory effects (Figure 2C). To further confirm the specificity of alpha-bands, we extracted the temporal dynamics within other frequency bands (delta, 1-2 Hz; theta, 3-6 Hz, beta, 15-25 Hz) in Experiment 1, none of which showed significant memory effects (Figure 2—figure supplement 3C). It is noteworthy that we used Experiment 1 as a pre-test to determine the neural marker for memory reactivations, and thus only assessed alpha-band temporal profiles in further experiments.

The spectrum results have been added (subsection “Experiment 1: Alpha-band memory effects (one-item memory)”, first paragraph; subsection “Correlation coefficient calculation and permutation test”, last paragraph; Figure 2—figure supplement 2C). The results within other frequency bands have been added (subsection “Experiment 1: Alpha-band memory effects (one-item memory)”, last paragraph; Figure 2—figure supplement 3C).

In order to draw the conclusions, it seems to me that each of these must be true. Please provide a more thorough explanation of the logic and assumptions behind using the TRF method to tag item reactivations and evidence that these assumptions are reasonable and supported by the data.

Again, we thank the reviewer for raising the important criticism and apologize for failing to make the underlying logic and assumptions clear in previous version. We have now provided a thorough explanation of all the logic and assumptions in the new version. The new control analysis results have also been added in supplementary figures (Figure 2—figure supplement 1C, Figure 2—figure supplement 2C).

2) All reviewers felt strongly trial-by-trial evidence for item maintenance should be presented. For the reported patterns to truly reflect a physiological pattern akin to replay, it would be important to demonstrate that individual electrodes/subjects sequentially represent each of the two different stimuli on individual trials. Otherwise, there is the concern that the results may be the effect of averaging such that the observed patterns were caused by different items, electrodes, or subjects showing activations at different latencies according to list position. I think it is vital that a true replay analysis be performed and the authors need to explain this analysis very clearly. The authors can satisfy this concern by ensuring that they report and document an analysis that is focused on identifying replay at the single trial level.

We fully agree with the reviewers that to essentially support the sequential reactivation profiles during maintenance, it is important to exclude the possibility that different electrodes, different subjects, and different items might elicit activations at different temporal latencies, and it might be the averaging that caused the observed sequential profiles. Replicating similar effects in single trials would thus exclude the alternative interpretation.

Meanwhile, we would like to point out a technical limitation about TRF approach that unfortunately could not be met in single-trial data in the current design. In particular, reliable estimation of TRF response would require neural signals with enough temporal length as well as the same duration of the corresponding stimulus sequences, from which the TRF response characterizing their linear relationship could be calculated. Previous study has proposed that neural signals of at least 120 secs would be needed to obtain TRF responses with reasonable SNR (Lalor et al., Brain Signal Analysis, 2009). Theoretically, dissociating multiple TRFs from brain signals would require even longer signals. Back to our studies, each trial was 5 s, and we only analyzed the middle 4 s to avoid influences from onset and offset response. Therefore, to obtain TRF responses for multiple probes (e.g., 2 TRFs in Experiment 1 and 3 TRFs in Experiment 2) from the same EEG signals, we would need at least more than 40 trials for the calculation. Single-trial analysis, unless it was designed to be very long, could not meet the signal length requirement. We have added the limitations of TRF approach in Materials and methods (subsection “TRF computation”, last paragraph).

Furthermore, we calculated the SNR of TRF responses for Experiment 1 and Experiment 2 as a function of signal length. As shown in Author response image 2, in both experiments, the SNR of TRF responses reached a steady level when the signal length was around 320 secs (equal to 80 trials). The overall SNR in Experiment 2 was much lower than that in Experiment 1, given that more TRF responses needed to be estimated in Experiment 2. The results further support that we would need adequate signal length (i.e., more trials here) for a reliable TRF estimation.

**Author response image 2. respfig2:** 

Although we could not perform single-trial TRF analysis, we fully agree the reviewer that it is critical to exclude the possibility that the results might be due to the combination of responses across subjects, channels and items, which might elicit distinct temporal activation patterns. We thus performed additional control analyses to address the three important concerns.

First, as shown in Figure 3D, we did a t-test on the normalized alpha-band time courses for the WM-TRF responses, demonstrating that the 2^nd^ -WM-TRF showed significant response around 80-100 ms, whereas the 1^st^ -WM-TRF elicited significant response around 260-300 ms. The results thus suggest that the observed backward sequential activations were consistent across individual subjects. Clarifications have been added in text (subsection “Experiment 2: Sequential alpha-band response for sequence memory”, third paragraph).

Second, we replotted the alpha power time courses for the 1st-WM-TRF and 2nd-WM-TRF for each of the ROI electrodes (Pz, P1, P2). As shown in Figure 3—figure supplement 1A, they revealed similar temporal profiles as that of the averaged pattern, supporting that the observed sequential activations were also consistent across ROI electrodes. New results and discussion have been added (subsection “Experiment 2: Sequential alpha-band response for sequence memory”, third paragraph; Figure 3—figure supplement 1A).

Finally, to examine whether different items would elicit distinct temporal patterns (e.g., red probes would be associated with responses with earlier latency, etc.), we replotted the TRF responses for different combinations of colors (i.e., red, blue, and green). Each time we removed one color condition and recalculated the TRF responses (we need enough trials for TRF estimation) for the remaining items. The logic is that if probes with different colors elicited different temporal patterns, we would expect to see inconsistent response profiles for different items. Meanwhile, as shown in Figure 3—figure supplement 1B, all the three conditions showed similar backward sequential activations, supporting that the observed sequential activations were also consistent across items, and were not biased by certain types of item. New results and discussion have been added (see the aforementioned subsection; Figure 3—figure supplement 1B).

Taken together, we thank the reviewer for raising the crucial concern. We have clarified why we could not perform single-trial TRF analysis (due to the technical requirement in TRF estimation) and instead needed sufficient number of trials (given the 4 s length per trial in our experiment) to obtain reliable TRF responses. Furthermore, to further address the concerns raised by the reviewers, we have performed several new control analyses. Our results demonstrate that the observed backward sequential profiles were not due to individual subject, or single channel, or combination of different items. In fact, we observed consistent response patters across subjects, channels, and items. We have added more clarifications and interpretation in Results (subsection “Experiment 2: Sequential alpha-band response for sequence memory”) and Materials and methods (subsection “TRF computation”). We have also added the new control analysis results in supplementary figures (Figure 3—figure supplement 1A, B).

3) Could the timing differences between the item reactivations be completely explained by the recency effect, or differences in memory strength by encoding position? For example, if we manipulated the sequence such that the item in the first position was always more salient than the item in the second position, would the temporal coding pattern during the maintenance period also flip (item 1 reactivated first, item 2 reactivated second)? A finding like this would suggest that the temporal reactivation differences are less about the encoding of order and more about the strength of encoding, or the 'attentional priority' of the items.

We appreciate the interesting alternative interpretation for our results. Specifically, the backward sequential pattern and its association with recency effect might be explained by the account that the better the memory strength or the attentional priority is, the earlier the latency would be. We thus performed a new control analysis in Experiment 2 by dividing subjects into two groups based on their memory accuracy instead of the recency effect and then examined the respective response profiles. If the ‘memory strength’ or ‘attentional priority’ interpretation were correct, we would expect to see temporal shift in the response pattern between the two groups given their different memory strength. In particular, the high-performance group would show earlier response compared to the low-accuracy group.

However, as shown in Figure 3—figure supplement 1C, high-accuracy (solid line) and low-accuracy (dashed line) groups did not show shift in temporal latency and actually elicited quite similar backward sequential responses as before. The new results thus exclude the possibility that it is the memory strength rather than the recency effect that leads to the backward sequential responses. Moreover and interestingly, we observed a trend that the high-accuracy group elicited stronger response than the low-accuracy group, consistent with findings in Experiment 1. The new control analysis and associated discussions have been added (Discussion, fifth paragraph; Figure 3—figure supplement 1C).

As to the interesting question about whether the saliency manipulation would change the order of the reactivation during maintenance, we believe it would depend on whether the recency behavior would be modulated correspondingly. If the 1^st^ item is far more salient than the 2^nd^ item, the recency effect probably would also be influenced and become weaker. As a result, it is quite possible that the reactivation sequence would be reversed. Nevertheless, it is noteworthy that in our experiments, items in the memory list were matched in both physical saliency (simultaneously presented and equally salient) and attentional priority (equally task-relevant) by presenting them in one display and employing sequential cuing to instruct subject to form sequence memory (Figure 3). We thank the reviewer for the interesting idea to test whether physical saliency manipulation could modulate the reactivation sequence during maintenance and believe further studies should be performed to fully address the issue.

Taken together, we appreciate the proposed interpretation for our results. We hope that our new control analysis could convince the reviewers that the observed backward sequential response could not be simply accounted for by the memory strength or attentional priority interpretation. We have added more interpretation and discussion in Discussion (fifth paragraph). We have also added the new control analysis results in Figure 3—figure supplement 1C.

[Editors’ note: the author responses to the re-review follow.]

The reviewers all agreed that the revision was very responsive and served to increase confidence in the results as well as clarified the methodological approach. However, there were still significant questions about the validity of the TRF approach especially given that the methods and analyses are so complex. Thus, as you can see, we reached out to another reviewer, an expert in the method. The comments of this reviewer echoed the concerns of the other reviewers and highlight the important need to validate the method with chromatic stimuli as suggested. Furthermore, for the reasons highlighted, this needs to be done on a single subject level. Because this would require new data collection and analyses, it is not clear how these analyses will turn out. However, if you are able to validate the paradigm with chromatic stimuli in the manner suggested by reviewer 4, as well as address all of the other concerns reviewer 4 brings up, we could consider a new submission at that time.

As summarized by the editor, the key concern raised by the reviewers is that we should provide more evidence validating the TRF approach with chromatic stimuli at single subject level. First, as suggested by reviewer 4, we have collected *new data* from additional subjects who were presented with a single chromatic disc stimulus. The results showed clear and robust TRF response at single subject level (see results as below), consistent with previous work (VanRullen and Macdonald, 2012) and our own study (Jia et al., 2017). Second, we have analyzed a new dataset in another study (N=13) that also used TRF to tag color probes, again confirming the approach.

That being done, we would like to point out that the skepticism raised by the reviewers might derive from some misunderstanding about our approach. We failed to make it clear enough in the previous manuscript for which we apologize. To clarify, we actually employed a well-established approach to track *luminance* sequence of color probes rather than the color change of the probes (the color of probes actually kept constant). Luminance modulation is widely used to tag item-specific brain response. For example, by applying different luminance modulation frequencies at different features (e.g., red or blue dot patterns), the corresponding steady state response (SSR) for different features could be dissociated to further examine attentional effects (Muller et al., PNAS, 2006). Here, rather than a sinusoidal luminance modulation in SSR, TRF approach employed a random luminance sequence to tag different color probes. To make the points more clear, we have modified the method description and added clarification, as well as adding a new figure to illustrate the TRF responses (new Figure 2—figure supplement 1). We have also made point-by-point responses and clarifications to other criticisms raised by reviewer 4 (see details in response to reviewer 4).

**Author response image 3. respfig3:** TRF responses for the single color probe experiment (as suggested by reviewer 4, two subjects).

**Author response image 4. respfig4:** TRF response for a new dataset in another experiment (N = 13).

We have appended the reviews below but please note that reviewers 1 and 3 made their comments as an initial response to the revision which was very responsive. However it was upon further consultation with these reviewers that it was clear we needed to consult with an expert on the TRF methods.Reviewer #3:I believe the authors adequately addressed most of the concerns raised. The only remaining issue is whether these results hold on a single-trial level. While I understand that the TRF (by definition) cannot be computed for single trials, I would have liked to see an intermediate solution: recompute the analysis for binned trials (i.e., smaller subsets). However, I do understand the SNR concerns raised, and it may have to remain for future work. Maybe the authors can at least add a couple sentences to the Discussion to that extent?

Thanks for the suggestion. Actually, we did the analysis on smaller subsets, as the reviewer suggested. As shown in Figure 3—figure supplement 1B, which plots the response profiles for different color probe combinations (a subset of all trials), and the results were consistent across different combinations.

Related to this, I did not understand how Figure 3D proves consistency over subjects? Because it is normalized? Maybe I am missing the point, but plotting the single subject results in addition to the grand average would have been more compelling.

Sorry for the confusion. Figure 3D plots the grand average results (mean ± SEM) and corresponding paired t-test results, based on which we claimed that the sequential profiles were consistent over subjects.

Reviewer #4:I have read the manuscript by Huang et al., the reviewing editor's summary of the reviewers' comments, and the authors' response to these comments. Although the authors extensively revised the manuscript based on the reviewer's comments, I agree with my predecessors that the assumptions that TRF approach allows to track memory activation of distinct stimuli is not sufficiently validated.To my knowledge, there has not been yet a single paper validating the TRF approach for color stimuli (only achromatic stimuli have been used by several groups so far, e.g. Lalor and VanRullen), thus the authors first need to provide evidence that color stimuli can elicit a reliable TRF. This can be done by presenting a single colored stimulus and comparing the observed TRF with that obtained by shuffling the pairing between luminance sequences and concurrently recorded EEG time series. Crucially, this has to be demonstrated on a single-subject level. The latter is particularly important, because periodicity of TRF for achromatic stimuli has been shown to differ across participants, and the peak frequency of TRF is highly correlated with individual alpha peak frequency (VanRullen and Macdonald, 2012). Thus, group average analyses of TRF time courses (as done here), unless demonstrated that between-subject variability in TRF was not significant, can be driven by the subjects that have higher prominent TRF amplitude.

We appreciate the reviewer’s thoughtful concerns. We agree with the reviewer that previous studies only used TRF approach on achromatic stimuli, and there seems to lack convincing evidence for color tracking in TRF response.

As suggested by the reviewer, we have collected *new data* from two subjects who were presented with a single chromatic disc stimulus. As shown in Author response image 3, the results showed clear and robust TRF response at single subject level, consistent with previous findings (VanRullen and Macdonald, 2012) as well as our own TRF studies (e.g., Jia et al., 2017). Furthermore, we analyzed a new dataset that also used TRF approach to tag color probes (N=13) and the results are illustrated in Author response image 4. All the results consistently support that TRF approach could be used to track chromatic stimuli at single-subject level.

Meanwhile, we would also like to point out that there might exist some misunderstanding about the approach we used in the current study. The confusion might derive from our unclear presentations in previous manuscript for which we apologize. To clarify, here we used TRF to track ongoing *luminance* change of the color probes rather than the color change of the probes (the color of the probes kept constant in our study). We believe the reviewer would agree with us that luminance modulation is a well established method to tag item-specific brain response. For example, by applying different luminance modulation frequencies at different features (red or blue dots), the corresponding steady state response (SSR) could be dissociated for different features to further examine feature-based attention (Muller et al., PNAS, 2006). Here, rather than a sinusoidal luminance modulation in SSR, TRF employed a random luminance sequence to tag different color probes. We thank the reviewer for raising the confusing concerns. We now have added more clarifications in main texts.

Furthermore, rather than averaging TRF time course across subjects (“group average analyses of TRF time courses (as done here)”) as the reviewer thought, we actually calculated the TRF time course and the corresponding alpha power profile in each subject separately before grand average (see the method in the second paragraph of the subsection 2 TRF computation”). Our subsequent memory effect (Figure 2D, Figure 3D) was also based on paired t-test, and thus the effect could not be driven by single subject as worried by the reviewer.

In addition, we agree with the reviewer that previous studies have demonstrated different alpha peak frequencies across individuals. However, here we extracted the alphaband (8-11 Hz) power profiles for each subject, an approach that could tolerate different peak frequencies.

Based on the example TRFs in Figure 1, TRFs for chromatic stimuli differ substantially from the previously reported TRFs for achromatic stimuli (e.g. VanRullen and Macdonald, 2012), which reverberate up to 1 sec. Moreover, given the previous reports on difference in alpha-band power for different color stimuli (e.g. Yoto et al., 2007; DOI: 10.2114/jpa2.26.373), it is likely that TRF for different color stimuli would also be different. However, it is an empirical question that needs to be addressed prior to using TRFs for chromatic stimuli to tag items held in memory.

We thank the reviewer for pointing out the unclear issue. Actually, the TRF responses obtained in the present study were quite consistent with previous ones (VanRullen and Macdonald, 2012), and the TRF response, especially the alpha echoes, were clear and robust at single subject level (see new Figure 2—figure supplement 1). The impression might derive from our previous rough and unclear illustration for which we apologize. To address the issue, we have added a new Figure 2—figure supplement 1, which plots the TRF waveforms, spectrum, as well as those for shuffled data. It is clear that our results largely replicated previous findings (VanRullen and Macdonald, 2012). Although the alpha echoes in our results did not last up to 1 sec as previous study showed, they were still quite similar to our own recordings in another study using achromatic stimuli (Jia et al., 2017).

The review also raised concerns about possible color effect on TRF response. This is an important question, but it is noteworthy that here we mainly focused on the memory effects and we have carefully counterbalanced the location and color effects. We have added more clarifications in the main texts.

It has been previously reported that the phases of two TRFs elicited by achromatic flickering stimuli on the left and right hemifield exhibit phase differences across many channels (Lozano-Doldevilla and VanRullen, DOI: https://doi.org/10.1101/190595). Thus, it is possible that the phase differences reported here are not the property of the items stored in memory, but rather reflects the differences in propagation of different TRFs across cortex. Thus, again it is important to validate the TRF approach for several chromatic stimuli and to demonstrate that the phase differences are not related to the location of the stimuli in the retinotopic space.

We appreciate the raised concern about possible location effect. First, as stated in the response to the last point, we mainly focused on the memory effects and have carefully counterbalanced the location of the probes. Second, we extracted the alpha power profiles for each condition and each channel, thus already discarding the phase information in the TRF responses. Finally, our control analysis demonstrated that probes at different locations showed similar memory-related temporal profiles (Author response image 5).

**Author response image 5. respfig5:** Alpha power time courses for left and right probe.

Removing grand-average TRF responses from condition-specific TRF responses defeats the purpose of subject- and condition-specific TRF (subsection “Normalization”), and more importantly creates artificial oscillatory pattern (Figure 3D).

We appreciate the concern. We do not agree that the observed sequential replay was due to the normalization procedure. First, we could see clear sequential activation pattern in the raw data (Figure 3C). Second, we did permutation test on the data, and the same normalization procedure was performed on the surrogate data as well. If the normalization introduced artificial sequential pattern, the data would not pass the test.

Moreover, the normalization was performed for each condition and in each subject separately, and therefore it would not remove subject- and condition-specific TRF, as worried by the reviewer.

The electrode selection procedure is arguably biased to find condition differences. The authors selected the electrodes of interest based on the significant differences in TRFs between the WM and NWM conditions (subsection "Channel-of-interest", as well as authors' response: "According to the suggestion, we first selected channels based on the EEG alpha power, from which we then seek channels showing significant memory effects in TRFs."), and then show that there are indeed differences in TRFs in all subsequent analyses. To be statistically appropriate, channel selection procedure cannot involve data from condition differences, and should only be selected based on condition average or other criteria that are orthogonal to the subsequent analyses.

We agree with the reviewer that the main purpose of Experiment 1 was to seek whether there were channels showing significant memory effects. Meanwhile, we do not agree with the reviewer that the ‘channel-of-interest’ was a biased selection. In Experiment 1, the “channel-of-interest” has passed statistical test after multiple-comparison correction across time and channels (subsection “Experiment 1: Alpha-band memory effects (one-item memory)”, first paragraph). In other words, we used a non-biased approach (statistical test after multiple comparison correction) to determine channels showing memory effects (“channel of interest”). In Experiment 2, we used channels that were independently established in Experiment 1 to examine response patterns for sequence memory, and we don’t think the selection was biased. We have added more clarifications about this point.

Overall, despite the interesting topic and clever experimental approach taken by the authors, the manuscript in its current form does not feel up to the empirical and analytic standards one typically associates with eLife.

We appreciate the reviewer’s insightful and careful criticisms and suggestions. We have added new data, as suggested by the reviewer, and a new dataset to validate the TRF approach on chromatic stimuli. We have also added a new Figure 2—figure supplement 1 to better illustrate the original TRF responses in our study. We hope our new data would convince the reviewer that TRF could be used to tag brain response for color probes and the TRF responses is consistent with previous findings and robust at single subject level. Moreover, based on the reviewer’s criticism, we are aware that some presentations in our previous manuscript were unclear and confusing and would introduce possible misunderstandings. We have modified and clarified several parts, emphasizing the luminance modulation point.

[Editors’ note: the author responses to the re-review follow.]

The appeal provided additional useful information that increased the confidence in your reported effects. However, there were still issues brought up by the reviewers that were misunderstood and/or not clearly addressed. These issues (outlined below) must be addressed in full in a revision in order for the paper to be acceptable either with data or added discussion.1) The additional information you provided about the counterbalancing of color and location was very useful. An additional control that would strengthen your argument would be to directly compare the preferred phase of the TRF for different color/location combinations. The appeal included an additional figure illustrating that there is no effect of location in the alpha power timeseries. If possible with the existing data, it would useful to see the same analysis by color. If there are no reliable differences in the alpha power time course by color, then this cannot be a likely explanation for the phase differences for the different items maintained in memory.

We thank the editors for the comments. Unfortunately, our previous dataset (Experiment 1) did not tag the color of the probes and therefore we could not perform the control analysis as we did for locations. The reason is that, as stated in our previous response to reviews, we mainly focused on the memory effect in the original experimental design, and have carefully counterbalanced color and locations (for the same reason, in the additional 13-subject dataset experiment as shown before, we did not tag the color either).

Meanwhile, to examine the color effect as asked by the editor, we acquired “new data” from an additional 4 subjects who were presented with a single chromatic disc stimulus (blue or red color) and analyzed the 6-subject data (adding the original 2 subjects, see our response to reviews in the last round) for the two color probes respectively. As shown in Author response image 6, the alpha power time course for the two colors displayed similar temporal profiles (Author response image 6).

To test whether the two colors elicited alpha response with different latencies or different power, we calculated the alpha-band peak latency and the spectrum for the two colors respectively. Neither the peak latency (Author response image 6) nor the alpha-band power (Author response image 6) showed significant difference (peak latency: Wilcoxon signed-rank test, *p* = 0.14; power: paired t-test, *p* = 0.13).

**Author response image 6. respfig6:** Color Control experiment (N = 6). Six subjects were presented with single color probes (blue or red), while performing a central fixation task. The TRF response was calculated for red and blue color probes respectively (Red-TRF and Blue-TRF). Each condition was ran for 64 trials. (**A**) Grand average alpha (8-12 Hz) power time courses (mean ± SEM) of blue-TRF and red-TRF. (**B**) Peak latency of the alpha-band response profiles of blue-TRF and red-TRF (Wilcoxon signed-rank test, *p* = 0.14). (**C**) Power spectrum of blue-TRF and red-TRF responses (alpha-band power: paired t test, *p* = 0.13).

Moreover, we would like to point out that we have done a control analysis in the last revision by assessing the response profiles for different color combinations in Experiment 2 (Figure 3—figure supplement 1D). The results showed similar sequential-backward profiles as found before, further confirming that color could not be an explanation for the observed sequential reactivation.

That being done, we totally agree that the possible influences of color on brain responses is an important question and needs to be systematically investigated in future studies.

2) Based on the new data, and re-analysis of another dataset, chromatic TRFs seem to be indeed oscillatory and comparable to the previously reported achromatic TRFs at the single-subject level. This is reassuring and adds credibility to the use of white-noise chromatic stimuli for tagging memory items. However, no information was included about the number of trials used in this control study to derive TFRs? Is it comparable to what has been used in the original study? Please add this information into the paper.

Thanks for the question. The number of trial for each color condition in the control recording was 64, comparable to that used in the original study. The information has been added in the figure legends.

3) Also, it is not clear whether the plots provided in the appeal letter come from a single-color stimulus (which color?) or average of 3 colors? Please make this clear in the paper. Thus, this leaves open the original question asked by reviewers – are different color TRFs comparable in amplitude and phase (the property of TRFs that the results of this study hinge on)? Please show the data for each color separately.

Thanks for the question and we are sorry for the lack of descriptions in the previous version. In the control recordings, we actually used the same colors as we did in Experiment 1 (red and blue probes) and the results previously shown were the average of the responses for the two colors. The information has been added in the figure legend. As to the color effect concern and the request for separate color plotting and comparison, please see our response to the 1^st^ comment.

4) Several statistical decisions remain unclear and need to be clearly motivated and described. For example, the paper stated "In Experiment 1, the "channel-of-interest" has passed statistical test after multiple-comparison correction across time and channels”. In other words, we used a non-biased approach (statistical test after multiple comparison correction) to determine channels showing memory effects ("channel of interest")." Thus, it appears that a multi-step procedure was used to select the channels of interest? The logic for this is not explained. Please explain the logic carefully and clearly. Although seemingly minor, why did you choose a p value of p =.058 instead of the conventional.05? (In the previous version reviewers also noted that details of a 'bootstrap' or 'permutation' test were also not clearly described). If you choose to submit a revised manuscript, please ensure that all statistics are clearly explained, motivated and reported. Otherwise, we cannot consider the paper suitable for publication.

Thanks for the comments regarding statistical descriptions. First, as to the channel-of-interest selection, the editor is correct that it was a multi-step procedure. Based on the prominent alpha-band activations in the TRF responses as observed, we first selected occipital channels showing significantly strong overall alpha activations (independent criteria). Among these channels, we then searched ones showing memory effect (WM－NWM alpha-band difference) after multiple-comparison correction across time and channels. The procedure has been described in the main texts (Results) as well as in Materials and methods (subsection “Channel-of-interest”) in previous version. To further clarify the logic as requested by the editors, we have added texts to explain why we first selected channels showing strong alpha responses.

Second, the reason for the seemingly strange 0.058 p-value was due to our original understanding of the journal policy that specific p value rather than the p threshold should be stated. Actually, when we did the channel-of-interest search based on the bootstrap test with multiple comparison correction, two channels (Pz, P2) passed the test with *p* <= 0.05, and three channels (Pz, P1, P2) passed the test for *p* <= 0.06 (marginally significant). We set the p threshold as 0.06 to incorporate more channels to increase the signal-to-noise ratio. We have now added the rationale in Materials and methods (subsection “Channel-of-interest”).

We have now carefully gone through all the statistical tests throughout the manuscript and have added more details for each test in Materials and methods.

5) A question was raised about whether the observed sequential replay was due to the normalization procedure. In the appeal, you noted 'First, we could see clear sequential activation pattern in the raw data (Figure 3C)." However, there is a concern that the normalization procedure itself (subtracting the average of the 3 conditions) seems to introduce an artificial oscillatory pattern. Looking at Figure 3C (non-normalized data), non-working memory item is arguably the only one that is oscillatory (most likely noise), and by removing the average of the three-curves one would end up with an oscillatory pattern in Figure 3D – the figure that summarizes the main finding of the paper. Please address this issue more directly by assessing whether there is indeed any evidence for oscillations in the raw data – without the normalization procedure.

We appreciate the concern and we are sorry for not correctly understanding the question by the reviewer. To directly assess the activation pattern in the raw data (i.e., without normalization), we directly compared the alpha response profiles for the 1^st^ and 2^nd^ memory-related items. As shown in Figure 3—figure supplement 1A, the 1^st^ -2^nd^ showed a clear negative-positive-negative oscillatory pattern. Further correlation analysis on the 1^st^ -2^nd^ responses supports the backward reactivation (negative correlation around lag of 200 ms) as well as the recurrent activation for the 2^nd^ item (positive correlation around lag of 400 ms). It is noteworthy that this analysis did not include the NWM items but showed the same results as before, thus excluding the interpretation that the observed backward-sequential reactivation was simply due to the subtraction of the NWM response. We have added the control analysis results in Figure 3—figure supplement 1 and main texts (subsection “Experiment 2: Sequential alpha-band response for sequence memory”, third paragraph).

Moreover, to directly assess the backward profile in the raw data, we extracted the alpha-band peak latency for the 1^st^ and 2^nd^ unnormalized TRF responses, and they showed significant difference (Wilcoxon signed-rank test, *p* = 0.036). The results further support that the observed backward reactivation was not due the involvement of the NWM item and was present in the raw data. We have also added the new results in the revised manuscript (see the aforementioned subsection).